# Squeeze the Soaked Sponge: Efficient Off-policy RFT for Large Language Model

**Jing Liang**[*1]    **Jinyi Liu**[*2]    **Yi Ma**[*3]    **Hongyao Tang**[†2]    **Yan Zheng**[†1]

**Shuyue Hu**[4]    **Lei Bai**[4]    **Jianye Hao**[†2]

[1]School of New Media and Communication, Tianjin University
[2]School of Computer Software, Tianjin University
[3]School of Computer and Information Technology, Shanxi University
[4]Shanghai Artificial Intelligence Laboratory

## Abstract

Reinforcement Learning (RL) has demonstrated its potential to improve the reasoning ability of Large Language Models (LLMs), yet most existing Reinforcement Finetuning (RFT) methods are inherently *on-policy* RL, failing to reuse historical data and thus preventing efficient scaling. In this work, we explore the potential of *off-policy* RL to leverage historical data for rollout-efficient RFT. Specifically, we propose **Re**incarnating **Mix**-policy Proximal Policy Gradient (**ReMix**), which enables on-policy RFT methods to leverage off-policy data. ReMix consists of three major components: (1) Mix-policy proximal policy gradient with an increased Update-To-Data (UTD) ratio that utilizes the data from both current and past policies for efficient training; (2) KL-Convex policy constraint that combines the KL constraints on the base and precedent model to balance stability and flexibility; (3) Policy reincarnation that replaces the base model with the mix-policy RFT model in the mid way of training and restarts on-policy training, to achieve a seamless transition from early efficiency to steady convergence. In our experiments, we train a series of ReMix models based on PPO, GRPO from 1.5B, 7B base models. On five math reasoning benchmarks (i.e., AIME'24, AMC'23, Minerva, OlympiadBench, and MATH500), ReMix achieves an average Pass@1 accuracy of **52.10%** (with **0.079M rollouts**) and **64.39%** (with **0.011M rollouts**) on 1.5B and 7B models, respectively. Compared with 15 recent advanced models, ReMix shows SOTA-level performance with an over **30x to 450x reduction in training cost in terms of rollout data volume**, demonstrating superior training efficiency. Additionally, our multifaceted analysis reveals insightful findings, including the implicit preference for shorter responses of off-policy RFT, the collapse mode of self-reflection under severe off-policyness, etc. The code and the trained models are available at https://anitaleungxx.github.io/ReMix/.

## 1 Introduction

The emergence of Large Language Models (LLMs) has lifted artificial intelligence to a next level, with the milestone works like (OpenAI, 2022; Jaech et al., 2024; Bai et al., 2022a; Trung et al., 2024; Guo et al., 2025). Consistent efforts are being made to push forward the limits of LLMs in performing deeper thinking and solving more complex tasks (Li et al., 2025b). Recently, Large Reasoning Models (LRMs) (Jaech et al., 2024; Guo et al., 2025; Kimi et al., 2025; Yang et al., 2025a) have taken the stage and attracted great attention, showing that a significant improvement of problem-solving ability can be achieved by a long human-like reasoning process (i.e., *slow thinking*), especially in scenarios like Math, Coding, Scientific Q&A, etc. One of the central recipes of LRMs is Reinforcement Finetuning (RFT) (Trung et al., 2024). By treating the LLM as a policy model, the LLM can follow the philosophy of Reinforcement Learning (RL) (Sutton & Barto, 1998) and learn to

---

[*]Equal contribution.

[†]Corresponding authors: `jianye.hao@tju.edu.cn, yanzheng@tju.edu.cn, tanghongyao@tju.edu.cn`.

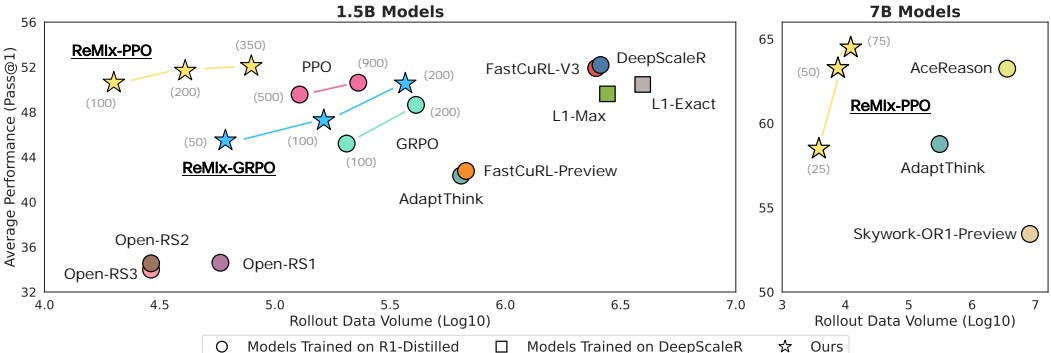

Figure 1: **Efficiency-performance comparison** for 1.5B models (*left*) and 7B models (*right*) in terms of Rollout Data Volume (i.e., total number of responses generated during training) v.s., Average Pass@1 Accuracy on five math reasoning benchmarks. An ideal model should appear in the top-left.

reason and answer the queries according to the reward signals, e.g., either from a verifiable reward function (Guo et al., 2025) or a learned reward model (Bai et al., 2022a).

Although RFT opens another space for more powerful reasoning ability of LLMs, the longstanding and notorious shortcoming of RL — *sample inefficiency* — still exists. In another word, RFT usually needs significantly more computational cost (e.g., rollouts and training) than SFT due to its *trial-and-error* nature. The inefficiency of RL poses a stringent bottleneck on time and cost, consequently preventing further scaling of model size and response length of LLMs. Currently, policy gradient algorithms like PPO (Schulman et al., 2017), GRPO (Shao et al., 2024), RLOO (Ahmadian et al., 2024) are widely adopted for RFT of LLMs due to their stable learning performance and friendliness to engineering. However, they are all *on-policy* algorithms, which are known to be sample inefficient as the data generated by the online policy is dropped after each iteration. In the literature of RL, *off-policy* algorithms are naturally more sample efficient since they also learns from the data generated by historical policies (i.e., *experience*) (Sutton & Barto, 1998; Silver & Sutton, 2025). Following this direction, recent research has begun to incorporate off-policy data in RFT in different ways, including using nonuniform replay strategies (Li et al., 2025a), learning from positive and negative signals asymmetrically (Roux et al., 2025; Arnal et al., 2025), proposing new learning objectives based on generation consistency (Tang et al., 2025; Cohen et al., 2025), and learning from demonstrations of superior models (Yan et al., 2025), etc. Despite the efforts made by these works, off-policy RFT remains underexplored in two aspects: (1) None of these methods was compared with SOTA models on multiple mainstream math reasoning benchmarks, leaving training efficiency and final performance of these methods untested thoroughly; (2) The influence of off-policy learning on the learning process of reasoning ability remains unknown, which impedes essential understanding of off-policy learning for RFT and advancement of effective methodologies.

In this paper, we study off-policy RL for post-training finetuning of LLMs, aiming to achieve SOTA-level reasoning ability efficiently and unbox the effects of off-policy learning for useful insights. We propose **Re**incarnating **Mix**-policy Proximal Policy Optimization (**ReMix**), a general approach to enable on-policy proximal policy gradient methods to leverage off-policy data efficiently. ReMix consists of three major components: (1) Mix-policy proximal policy gradient with an increased Update-To-Data (UTD) ratio (Chen et al., 2021) leverages the data generated by both current policy and past polices for efficient training; (2) KL-Convex policy constraint (Ma et al., 2024b) combines the KL constraints on the base model and the precedent model to balance the trade-off between stability and flexibility during training; (3) Policy reincarnation (Agarwal et al., 2022) replaces the base model with the mix-policy RFT model in the mid way of training and restarts on-policy training, to achieve a seamless transition from efficient early-stage learning to steady asymptotic improvement. Under the synergy of the three components, ReMix is able to improve the reasoning ability of LLMs efficiently while retaining a stable and flexible training process.

In our experiments, we adopt PPO and GRPO as representative on-policy methods and implement **ReMix-PPO** and **ReMix-GRPO**. We use `DeepSeek-R1-Distill-Qwen-1.5B` and `-7B` (Guo et al., 2025) as the base models, and train our models based on DeepScaleR-Preview-Dataset (Luo et al., 2025). We conduct a range of comparative evaluations against 15 recent advanced models on five math reasoning benchmarks, including AIME'24, AMC'23, Minerva (Lewkowycz et al., 2022),

OlympiadBench (He et al., 2024), and MATH500 (Hendrycks et al., 2021). Figure 1 summarizes the experimental results in a view of efficiency-performance comparison. Our method achieves an **average Pass@1 accuracy of 52.10%** (for 1.5B model) with **0.079M response rollouts, 350 training steps** and achieves **63.27%/64.39%** (for 7B model) with **0.007M/0.011M response rollouts, 50/75 training steps** respectively, showing SOTA-level performance and an over **30x to 450x training cost reduction in terms of rollout data volume**. Moreover, to gain a better understanding of off-policy learning for RFT, we conduct multifaceted studies and analysis, revealing insightful findings including the implicit preference for shorter responses due to the Whipping Effect of off-policy discrepancy, the collapse mode of self-reflection behavior under the presence of severe off-policyness, the performance under response length constraint, the impact of prompt format, etc.

## 2 PRELIMINARIES

**Reinforcement Learning for LLM Fine-tuning**  Reinforcement Fine-Tuning (RFT) is a paradigm for adapting pre-trained LLMs to specific downstream tasks using RL (Trung et al., 2024; Jaech et al., 2024). In this paradigm, text generation is modeled as a Markov Decision Process (MDP) $M = (\mathcal{S}, \mathcal{A}, P, R, \gamma)$, where a **state** $s_t = (q, y_{1:t}) \in \mathcal{S}$ is the prompt with the output generated so far, and the **action** $a_t \in \mathcal{A}$ is the next token selected from the vocabulary $\mathcal{V}$. Hence, the **transition** $P(s_{t+1}|s_t, a_t)$ is deterministic in this context. An episode start from a prompt $s_0$ (out of a predefined set $\mathcal{D}_0$) and terminate at an end-of-sequence token or by the maximum sequence length $H$.

The **reward** $R(s_t, a_t)$ signal is issued by either a rule-based reward function or a learned reward model. In the scope of this paper, we consider the verifiable reward function. For any non-terminal timestep $t < T - 1$, $R(s_t, a_t)$ is 0; on completion, the terminal reward, denoted by $R(\tau)$ for the whole sequence, equals 1 if $\tau$ produces a correct and well-formatted answer and 0 otherwise. The **policy** $\pi_\theta(a_t \mid s_t)$ in the MDP is the LLM itself, parameterized by $\theta$, and it defines a probability distribution of next-token generation. We use $d_\tau^{\pi_\theta}$ to denote the distribution of the output sequence $\tau$ generated by $\pi_\theta$ and use $d_{s,a}^{\pi_\theta}, d_s^{\pi_\theta}$ for the state-action pairs $(s, a)$ and the state respectively. The learning objective of an RL policy is to maximize the reward function, i.e., $\pi^* = \arg\max_{\pi_\theta} J(\pi_\theta)$.

**Proximal Policy Gradient Methods for RFT**  Proximal Policy Optimization (PPO) (Schulman et al., 2017) is a canonical Policy Gradient (PG) method (Sutton & Barto, 1998) to maximize $J(\pi_\theta)$, which offers stable training and implementation simplicity. PPO is further developed to be GRPO (Shao et al., 2024) with a group-based advantage estimator. The policy optimization objective of PPO is formulated as:

$$L^{\text{CLIP}}(\theta) = -\mathbb{E}_{s,a \sim d_{s,a}^{\pi_{\theta_{\text{old}}}}} \left[ \min \left( r_\theta(s, a)\hat{A}(s, a), \text{clip}(r_\theta(s, a), 1 - \epsilon, 1 + \epsilon)\hat{A}(s, a) \right) \right], \quad (1)$$

where $r_\theta(s, a) = \frac{\pi_\theta(a|s)}{\pi_{\theta_{\text{old}}}(a|s)}$ represents the importance sampling ratio between the current policy $\pi_\theta$ and the old policy $\pi_{\theta_{\text{old}}}$ (i.e., the policy before the update), $\hat{A}(s, a)$ is an estimator of the advantage function with GAE (Schulman et al., 2016) as a popular choice, and the clip ratio $\epsilon$ defines the clipping range that determines the proximity of policy update, thereby enhancing stability. When applying RL for LLM, a KL-divergence penalty is often added to prevent the policy from deviating too far from a reference model $\pi_{\text{base}}$, e.g., the SFT model. The complete objective is:

$$L^{\text{PPO}}(\theta) = \mathbb{E}_{s \sim d_{s,a}^{\pi_{\theta_{\text{old}}}}} \left[ L^{\text{CLIP}}(\theta) + c\mathcal{H}[\pi_\theta](s) \right] + \beta \cdot \underbrace{\mathbb{E}_{s \sim d_s^{\pi_{\theta_{\text{old}}}}} \left[ D_{\text{KL}}(\pi_\theta(\cdot \mid s)||\pi_{\text{base}}(\cdot \mid s)) \right]}_{L_{\text{KL}}(\theta; \pi_{\text{base}})}, \quad (2)$$

where $\mathcal{H}[\pi_\theta](s)$ is the entropy of the policy $\pi_\theta$ at state $s$, $D_{\text{KL}}$ is the KL metric, and $c, \beta$ are weighting coefficients. In this work, we view both PPO, GRPO, and other variants of PPO as Proximal Policy Gradient (PPG) methods.

## 3 REINCARNATING MIX-POLICY PROXIMAL POLICY OPTIMIZATION

In this section, we introduce our method, Reincarnating Mix-policy Proximal Policy Optimization (ReMix), for efficient and stable RFT of LLMs. Specifically, ReMix consists of three synergistic innovations: (1) Mix-policy proximal policy gradient with an increased Update-To-Data (UTD) ratio for efficient training (Section 3.1); (2) KL-Convex policy constraint to balance stability and flexibility (Section 3.2); (3) Policy Reincarnation for a smooth transition from efficient early learning to stable asymptotic improvement (Section 3.3). We introduce the three components below.

### 3.1 MIX-POLICY PROXIMAL POLICY GRADIENT WITH INCREASED UTD RATIO

While proximal policy gradient methods like PPO, GRPO deliver strong performance in RFT, the on-policy nature of these methods leads to a significant bottleneck on data utilization. To address this inefficiency, we trace back to the off-policy RL literature. To be specific, we revisit the generalized proximal gradient theory (Queeney et al., 2021), which allows proximal gradient methods to make use of historical trajectories generated during the past policy optimization process.

In this work, we launch the renaissance of off-policy RL for RFT and introduce an On-/Off-policy *Mixed* Proximal Policy Gradient method (**Mix-PPG**) that strategically leverages both off-policy and on-policy data within a unified objective function. Formally, for policy at training step $k$, the mini-batch training data are sampled from a mixture of sources: the trajectories generated by historical policies (i.e., $\pi_{k-i}$ for $i \sim \nu$), and the trajectories of the current policy (i.e., $\pi_k$). This hybrid sampling strategy balances two competing purposes: (1) Data Reuse: Exploiting past trajectories reduces the autoregressive rollout and inference overhead; (2) Distribution Alignment: Maintaining sufficient on-policy samples prevents training instability and degradation due to the divergence from the current state-action distribution. The policy optimization objective function can be formalized as:

$$L_k^{\text{Mix-PPG}}(\theta) = -\mathbb{E}_{i \sim \nu}\Big[\mathbb{E}_{(s,a) \sim d_{s,a}^{\pi_{k-i}}} \min\Big(r_\theta^{k-i}(s,a) A^{\pi_k}(s,a),$$
$$\text{clip}\Big(r_\theta^{k-i}(s,a), \frac{\pi_k(a \mid s)}{\pi_{k-i}(a \mid s)} - \epsilon, \frac{\pi_k(a \mid s)}{\pi_{k-i}(a \mid s)} + \epsilon\Big) A^{\pi_k}(s,a)\Big)\Big], \quad (3)$$

where $i \sim \nu$ with $i \in \{0, 1, \ldots, N\}$ is a combined distribution over historical policy indices $\pi_{k-i}$ and the current policy $\pi_k$ (i.e., when $i = 0$), the importance sampling ratio $r_\theta^{k-i}(s,a) = \frac{\pi_\theta(a|s)}{\pi_{k-i}(a|s)}$. Notably, we incorporate a sampling strategy to strike a balance between training stability and efficient data utilization by using a portion $p$ of off-policy data drawn from $\pi_{k-i}$ and $1 - p$ on-policy data drawn from $\pi_k$ with $p \in [0, 1]$. A theoretical discussion on $\nu$ can be found in Appendix D.2. Now, we are ready to replace the on-policy policy optimization objective, e.g., the $L^{\text{Clip}}(\theta)$ term in Eq. 2, with the Mix-PPG objective $L_k^{\text{Mix-PPG}}(\theta)$ for efficient data utilization. One thing to note is, we found that explicitly maintaining the portion of on-policy data at a sufficient level is critical to effective training, as much off-policyness data will lead to a degradation or even collapse (as depicted in Figure 2).

To further improve sample efficiency, we leverage an increased Update-To-Data (UTD) ratio (Chen et al., 2021), defined originally as the number of gradient updates per environment interaction step. Specifically, we use a UTD ratio $m$, i.e., performing repeated gradient updates on sampled data batches for $m$ times, thereby further reducing fresh environment interaction demands.

### 3.2 KL-CONVEX POLICY CONSTRAINT

Conventional RFT imposes a static KL-constraint regularization on deviations from the base pre-trained model $\pi_{\text{base}}$. This rigid static constraint fails to accommodate evolving policy distributions, which could lead to suboptimal updates during the dynamic learning process.

Inspired by the recent study (Ma et al., 2024b), we propose to dynamically update the anchor objective to a convex combination of $\pi_{k-1}$ and $\pi_{\text{base}}$. On the one hand, by constraining the policy within the support of $\pi_{\text{base}}$, we enforce behavioral consistency with foundational capabilities, thereby preventing catastrophic forgetting of core skills. On the other hand, the constraint imposed on $\pi_{k-1}$ serves as a dynamic adaptation to the policy's current knowledge frontier. It facilitates iterative refinement of the policy and enables the policy to continuously evolve and improve steadily. With this mechanism, the policy can leverage the strengths of both the pre-trained model and the iterative refinement process.

Specifically, we reconcile the KL-constraint in RFT via a KL-convex policy constraint (**KLC**), which modifies the essential optimization objective described in Eq. 2 by replacing the conventional $L_{\text{KL}}(\theta; \pi_{\text{base}})$ with the KL-convex constraint term as formulated below:

$$L_{\text{KLC}}(\theta; \pi_{\text{base}}, k) = \mathbb{E}_s \big[\lambda D_{\text{KL}}\big(\pi_\theta(\cdot|s) \parallel \pi_{\text{base}}(\cdot|s)\big) + (1 - \lambda) D_{\text{KL}}\big(\pi_\theta(\cdot|s) \parallel \pi_{k-1}(\cdot|s)\big)\big], \quad (4)$$

where $\lambda \in [0, 1]$ balances base-model alignment and behavioral consistency with recent policy $\pi_{k-1}$. This convex combination preserves foundational capabilities while enabling targeted adaptation, acting as a conservative regularizer against over-specialization. We provide more theoretical discussion on KLC loss in Appendix D.3. Empirically, we found that using a decaying $\lambda$ consistently beat a

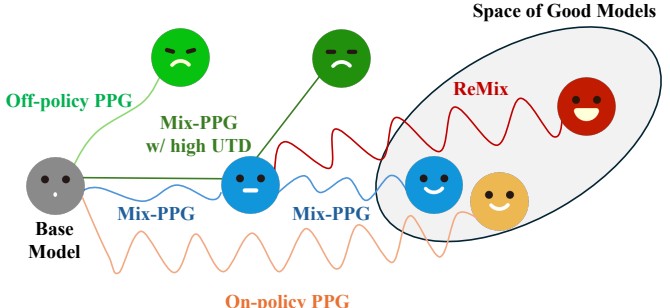

Figure 2: **The conceptual illustration of RFT for LLMs with different proximal policy gradient (PPG) methods.** Starting from a base model, (1) on-policy PPG methods (e.g., PPO, GRPO) train stably, yet uses data inefficiently. (2) Off-policy PPG is data-efficient. However, naively adopting it leads to a collapse. (3) To strike a balance, we introduce Mix-PPG, which boosts early-stage performance but still faces a slow asymptotic improvement and even a collapse when adopting a high UTD ratio. (4) Therefore, we propose policy reincarnation and introduce **ReMix**, which achieves better efficiency at no compromise of final performance.

fixed one. A linear decay schedule works well while other forms (e.g., exponential decay) did not make a significant difference.

### 3.3 POLICY REINCATENATION

While the mix-policy proximal PG method proposed above accelerates early-stage training, the off-policy bias in it can inevitably limit the asymptotic performance, as the empirical evidence later shown in Figure 4. Inspired by Reincarnating RL (Agarwal et al., 2022), we propose Policy Reincarnation in the context of RFT for LLMs, to seamlessly combine the advantage of both off-policy RL and on-policy RL, thus being more efficient at no cost of asymptotic performance.

To be specific, the training process consists of the Mix-PPG stage and the reincarnating on-policy PPG stage. First, the initial policy model is trained for a predetermined $T$ steps of gradient update according to the proposed Mix-PPG algorithm for quick improvement of policy performance. Thereafter, the reincarnation happens through two changes to the training setting: (1) reset the base model from the initial reference model $\pi_{\text{base}}$ to the current policy model $\pi_T$ (which alters the conventional KL constraint term), and (2) switch Mix-PPG to a on-policy PPG method (e.g., PPO or GRPO).

Finally, by composing Mix-PPG (Eq. 3), KL-convex policy constraint (Eq. 4), and policy reincarnation, we arrive at the complete method proposed in this paper, i.e., Reincarnating Mix-policy Proximal Policy Optimization (**ReMix**), as follows:

$$L^{\text{ReMix}}(\theta) = \begin{cases} \mathbb{E}_{d_{s,a}^{\pi_\theta}}\left[L^{\text{Mix-PPG}}(\theta) + c\mathcal{H}[\pi_\theta](s)\right] + \beta \cdot L_{\text{KLC}}(\theta; \pi_{\text{base}}, t) & \text{if } t \leq T \text{;} \\ \mathbb{E}_{d_{s,a}^{\pi_\theta}}\left[L^{\text{PPO}}(\theta) + c\mathcal{H}[\pi_\theta](s)\right] + \beta \cdot L_{\text{KLC}}(\theta; \pi_T, t) & \text{otherwise.} \end{cases} \quad (5)$$

Note that $t$ is the number of batch training steps and the two changes that occur upon policy reincarnation are highlighted in blue and red respectively. In Eq. 5, we use PPO as the on-policy PPG method for demonstration. For the case of GRPO, one can replace the advantage estimation in both $L^{\text{Mix-PPG}}$ and $L^{\text{PPO}}$ with the group-based estimation, as done in our experiments.

The efficacy of ReMix is two-fold. First, it leverages the advantages of Mix-PPG and on-policy PPG in boosting early-stage training and stable asymptotic improvement respectively, by establishing a seamless transition between the two stages. Second, the KL-convex policy constraint and the reset of the base reference model for KL constraint (i.e., $\pi_{\text{base}} \to \pi_T$) upon policy reincarnation offers a dynamics and looser constraint compared to the conventional static KL constraint, allowing fast policy training and a larger policy optimization space. For an intuitive understanding, we provide a conceptual illustration of RFT with different proximal PG methods in Figure 2.

## 4 EXPERIMENTS

In this section, we empirically evaluate the efficacy of ReMix on a range of commonly adopted Math reasoning benchmarks in terms of both accuracy and efficiency (Section 4.2), along with ablation

studies (Section 4.3) and multifaceted analyses (Section 4.4 and Appendix L). In addition, we provide extensive evaluation regarding more evaluation metrics, code generation, base RFT algorithms, etc.

## 4.1 EXPERIMENTAL SETUP

**Training** We use `DeepSeek-R1-Distill-Qwen-1.5B` and `-7B` (Guo et al., 2025) as the base models in our experiments. For implementation, we adopt PPO and GRPO as two representative on-policy proximal PG methods in our experiments, resulting in **ReMix-PPO** and **ReMix-GRPO**.

We use DeepScaleR-Preview-Dataset (Luo et al., 2025), which comprises approximately 40,000 unique problem-answer pairs sourced from AIME (1984–2023), AMC (prior to 2023), the Omni-MATH dataset (Gao et al., 2025), and the Still dataset (Min et al., 2024). We use the DeepScaleR's prompt format by default, instructing the LLM to follow structured step-by-step reasoning and produce a verifiable \boxed{} final answer. Full templates and instances are provided in Appendix G.

Our experiments are conducted using the `verl`[1] framework and the codebase derived from `tinyzero`[2]. For ReMix, we use an off-policy data portion $p = 0.4$, a UTD ratio $m = 2$, a historical policy window size $N = 2$, and we set the policy reincarnation step point to $T \in \{50, 100\}$ for ReMix-PPO and $T = 50$ for ReMix-GRPO. The KL-Convex coefficient $\lambda$ decays with training steps $t$ as $\lambda(t) = \max(1 - 0.1 \cdot \lceil \max(t - 50, 0)/10 \rceil, 0.5)$. We use these configurations by default, except in hyperparameter analysis. Prompts are truncated to 766 tokens, and the maximum generation length is **8,192** tokens. The detailed hyperparameter choices are presented in Table 8.

**Evaluation** We evaluate the performance of different models on a series of mathematical reasoning benchmarks, including AIME'24, AMC'23, Minerva (Lewkowycz et al., 2022), OlympiadBench (He et al., 2024), and MATH500 ((Lightman et al., 2023)) (None of these datasets are contained in our train set). During evaluation, we feed the entire context into the evaluation function. The models in comparison use the same generation settings as in training, except the `do_sample` parameter is set to `false`, resulting in greedy decoding. For the evaluation of baseline methods, we use the officially released checkpoints from HuggingFace to ensure fair results; for our models, we use the best checkpoints obtained within a specific training step budget, e.g., ReMix-PPO (200 Steps).

In our experiments, we focus on the evaluation of ReMix in terms of both **model performance** and **training efficiency**. For model performance, we mainly use **Pass@1** accuracy, and Avg@32 for small datasets (AIME'24 and AMC'23) in Appendix J.1. For training efficiency, we evaluate the models mainly in terms of *rollout data volume*, defined as the total number of rollouts generated by the model during training, which is usually **the dominant source** of computational cost during training in practice. We also use *training steps* (i.e., the number of rollout prompt batches) and *training duration* (i.e., the actual elapsed wall-clock time) as additional aspects for efficiency evaluation.

The detailed introduction of the compared baselines is provided in Appendix F. We also provide the discussion on related off-policy methods that are infeasible to compare with in Appendix F.3. For other training details, please refer to Appendix I.

## 4.2 PERFORMANCE EVALUATION FOR MATH REASONING

The performance evaluation in terms of Pass@1 accuracy on five math reasoning benchmarks are shown in Table 1 and Table 2, our method ReMix achieves consistent and substantial improvements over the base 1.5B/7B model on all five benchmarks. For **ReMix-PPO**, it achieves an **average performance gain** of **14.52 points** and **12.31 points** over 1.5B and 7B base models respectively, **achieving the second-best average score for 1.5B and the best for 7B among all the baselines**. In addition, compared with PPO (900 Steps, 1.5B) and PPO (200 Steps, 7B), our model achieves higher average scores within 100 steps for 1.5B and 50 steps for 7B. Similarly, our model exceeds GRPO (100 Steps, 1.5B) and GRPO (200 Steps, 1.5B) within 50 and 200 training steps, respectively. This indicates that ReMix is able to achieve competitive reasoning ability efficiently with overall no compromise in accuracy and even showing a higher accuracy.

More importantly, we move on to the evaluation in terms of training efficiency. This is shown in the last volume (i.e., Cost) of Table 1 and 2, and notably, Figure 1 illustrates **the efficiency–accuracy**

---

[1] https://github.com/volcengine/verl
[2] https://github.com/Jiayi-Pan/TinyZero

Table 1: **Pass@1 accuracy (%) and training cost (in terms of Rollout Data Volume) of 1.5B models.** **Bolded** and underlined values denote the highest and the second-highest scores in each dataset (i.e., column). '-' denotes that not enough information was found. **ReMix achieves better average scores than both the standard PPO and GRPO in a significantly more efficient manner**.

| Model | AIME'24 | AMC'23 | MATH500 | Minerva | Olympiad | Avg. | Cost |
|---|---|---|---|---|---|---|---|
| R1-Distill-Qwen-1.5B (Base) | 33.33 | 43.37 | 67.40 | 16.54 | 27.26 | 37.58 | N/A |
| Open-RS1 | 23.33 | 42.17 | 64.20 | 16.18 | 27.11 | 34.60 | 0.058M |
| Open-RS2 | 16.67 | 45.78 | 65.00 | 18.38 | 26.96 | 34.56 | 0.029M |
| Open-RS3 | 16.67 | 44.58 | 67.60 | 15.64 | 25.48 | 33.99 | 0.029M |
| AdaptThink | 13.33 | 57.83 | 78.60 | 23.90 | 38.07 | 42.35 | 0.643M |
| II-Thought | 26.67 | 56.63 | 73.00 | 23.16 | 40.89 | 44.07 | - |
| FASTCuRL-preview | 26.67 | 60.24 | 74.20 | 20.22 | 32.59 | 42.78 | 0.676M |
| FASTCuRL-V3 | 36.67 | 66.27 | 84.40 | 28.67 | 43.56 | 51.91 | 2.478M |
| L1-Exact* | 23.33 | **71.08** | 84.00 | 29.41 | 44.59 | 50.48 | 3.953M |
| L1-Max* | 20.00 | 69.88 | 83.00 | 29.04 | **46.37** | 49.66 | 2.764M |
| DeepScaleR | 40.00 | 65.06 | 83.20 | 29.04 | 43.41 | **52.14** | 2.519M |
| GRPO (100 Steps) | 30.00 | 56.63 | 75.80 | 25.37 | 38.22 | 45.20 | 0.205M |
| GRPO (200 Steps) | 36.67 | 61.45 | 80.00 | 25.37 | 39.70 | 48.64 | 0.410M |
| **ReMix-GRPO (50 Steps)** | 23.33 | 57.83 | 80.40 | 26.10 | 39.70 | 45.47 | 0.061M |
| **ReMix-GRPO (100 Steps)** | 23.33 | 62.65 | 82.00 | 28.68 | 39.70 | 47.27 | 0.163M |
| **ReMix-GRPO (200 Steps)** | 33.33 | 65.06 | **84.60** | 26.10 | 43.55 | 50.53 | 0.368M |
| PPO (500 Steps) | 36.67 | 62.65 | 82.60 | 25.73 | 40.14 | 49.56 | 0.128M |
| PPO (900 Steps) | 30.00 | 69.88 | 84.00 | 25.74 | 43.41 | 50.61 | 0.230M |
| **ReMix-PPO (100 Steps)** | 43.33 | 63.86 | 79.60 | 26.84 | 39.41 | 50.61 | 0.020M |
| **ReMix-PPO (200 Steps)** | **46.67** | 62.65 | 82.20 | 26.10 | 40.74 | 51.67 | 0.041M |
| **ReMix-PPO (350 Steps)** | $36.67^{\uparrow 3.34}$ | $69.88^{\uparrow 26.51}$ | $82.00^{\uparrow 14.60}$ | $30.15^{\uparrow 13.61}$ | $41.78^{\uparrow 14.52}$ | $52.10^{\uparrow 14.52}$ | 0.079M |

**trade-off** in terms of **rollout data volume** ($\log_{10}$ **scale**) versus **average Pass@1 accuracy**, where the scores are out of Table 1 and 2 (i.e., **Avg.** and **Cost**). In the ideal case, the model should appear in the top-left corner of the plot. To ensure a fair comparison, the rollout data volume of square-marked models (which means the models fine-tuned upon `DeepScaleR`) includes the data cost of training `DeepScaleR` itself. For ReMix-GRPO and GRPO, we report results after 200 training steps due to computational resource constraints.

Specifically, for 1.5B models, **ReMix-PPO** matches **DeepScaleR**, the strongest competitor, with just 0.079M vs. 2.519M rollouts, over **a 30× reduction** in rollout data volume. Also, **ReMix-PPO** reaches 50.61 after 0.020M rollouts, **10× fewer** than **PPO** (50.61 at 0.230M), highlighting rapid early gains. For 7B models, **ReMix-PPO** topping **AceReason-Nemotron** with over **a 450× reduction** in rollout data volume, and outperforming **PPO** with **a 6× reduction**. Notably, the average rollout response length of ReMix is lower than the baseline models (see Fig. 4), hence the exact efficiency should be higher. The corresponding detailed factors associated with computational cost for training all compared models above are shown in Appendix I.

> **Takeaway 1. ReMix can learn strong reasoning ability in a highly efficient way.**
>
> ReMix achieves SOTA-level accuracies at 1.5B and 7B scales on five math reasoning benchmarks, with an over 6x to 10x reduction in rollout data volume when outperforming PPO and an over 30x to 450x reduction when performing on par with (or exceeding) the best baseline.

## 4.3 ABLATION STUDIES

To assess the contribution of each components in ReMix, we conduct ablation studies focusing on both training dynamics and final performance. We use ReMix-PPO for the ablation studies.

The results of the ablation studies regarding Pass@1 accuracy are presented in Table 3. First, when Mix-PPG, the core of ReMix, is ablated, the method degenerates to PPO since it does not make sense any longer to apply other components of ReMix. Removing any single component among increased UTD, KL-Convex, and policy reincarnation leads to a final average score comparable to PPO but lower than ReMix within 500 training steps, Dropping policy reincarnation hurt the most among the three ablations. Further, when Mix-PPG works solely, it leads to an even lower score.

This reflects the off-policy nature in Mix-PPG: although it significantly increases the training efficiency, the off-policyness bias may hinder the convergence performance. The superiority in efficiency brought by Mix-PPG can be observed by referring to the first subplot of Figure 4: Mix-PPG shows a somewhat surprising boost of Pass@1 accuracy within the first 100 training steps, which an increased

Table 2: **Pass@1 accuracy (%) and training cost (in terms of Rollout Data Volume) of 7B models. ReMix-PPO achieves the best average score within 75 training steps**.

| Model | AIME'24 | AMC'23 | MATH500 | Minerva | Olympiad | Avg. | Cost |
|---|---|---|---|---|---|---|---|
| R1-Distill-Qwen-7B (Base) | 33.33 | 68.68 | 83.80 | 30.15 | 44.44 | 52.08 | N/A |
| ReasonFlux-F1 | 20.00 | 54.22 | 77.20 | 29.04 | 37.04 | 43.50 | - |
| Light-R1 | 30.00 | 66.27 | 87.00 | 34.56 | 47.56 | 53.08 | - |
| Skywork-OR1-Preview | 43.33 | 63.86 | 84.40 | 29.41 | 46.22 | 53.44 | >8.192M |
| Polaris | 40.00 | 63.86 | 87.60 | 36.40 | 48.00 | 55.17 | - |
| AdaptThink | 46.67 | 75.90 | 87.60 | 33.46 | 50.22 | 58.77 | 0.307M |
| AceReason-Nemotron | 60.00 | **80.72** | 89.00 | 36.40 | 50.07 | 63.24 | >3.584M |
| **ReMix-GRPO (75 Steps)** | 63.88 | 90.60 | 80.72 | 40.07 | 53.78 | 65.81 | 0.046M |
| **ReMix-GRPO (200 Steps)** | 64.37 | 91.60 | 81.93 | 39.34 | 53.19 | 66.09 | 0.163M |
| PPO (50 Steps) | 33.33 | 71.08 | 87.20 | 36.03 | 48.00 | 55.13 | 0.013M |
| PPO (100 Steps) | 40.00 | 77.11 | 90.00 | 35.66 | 51.56 | 58.87 | 0.026M |
| PPO (200 Steps) | 53.33 | 78.31 | 87.00 | 34.19 | 48.88 | 60.34 | 0.051M |
| **ReMix-PPO (25 Steps)** | 36.67 | 78.31 | 89.00 | 38.24 | 50.22 | 58.49 | 0.003M |
| **ReMix-PPO (50 Steps)** | 56.66 | 79.52 | 88.60 | **38.97** | 52.59 | 63.27 | 0.007M |
| **ReMix-PPO (75 Steps)** | 63.33$^{\uparrow 30.00}$ | 78.31$^{\uparrow 9.63}$ | 90.20$^{\uparrow 6.40}$ | 37.50$^{\uparrow 7.35}$ | 52.59$^{\uparrow 8.15}$ | 64.39$^{\uparrow 12.31}$ | 0.011M |

Table 3: **Ablation studies regarding Pass@1 accuracy.** The three components of ReMix work in synergy for both efficiency and final performance.

| Model | AIME'24 | AMC'23 | MATH500 | Minerva | Olympiad | Avg. |
|---|---|---|---|---|---|---|
| R1-Distill-Qwen-1.5B (Base Model) | 33.33 | 43.37 | 67.40 | 16.54 | 27.26 | 37.58 |
| PPO (500 Steps) | 36.67 | 62.65 | **82.60** | 25.73 | 40.14 | 49.56 |
| **ReMix-PPO (350 Steps)** | 36.67 | **69.88** | 82.00 | **30.15** | 41.78 | **52.10** |
| ReMix-PPO w/o UTD | 36.67 | 62.65 | 82.20 | 28.68 | **42.96** | 50.63 |
| ReMix-PPO w/o KL-Convex | 30.00 | 65.06 | 81.60 | 27.94 | 42.22 | 49.36 |
| ReMix-PPO w/o Policy Reincarnation | 20.00 | 67.47 | 82.00 | 26.84 | 40.00 | 47.26 |
| ReMix-PPO w/o UTD, KL-Convex, Policy Reincarnation | **40.00** | 57.83 | 80.40 | 25.74 | 39.55 | 48.70 |

UTD further enhances it; while the KL-Convex and policy reincarnation in ReMix contribute to the steady asymptotic improvement.

> **Takeaway 2. The three components work in synergy for efficiency and final performance.**
>
> Mix-PPG with an increased UTD boosts early-stage training significantly, while policy reincarnation plays a critical role to ensure asymptotic improvement.

**Training Curves** In addition to the efficiency evaluation in terms of rollout data volume, we present the training curves for ReMix-PPO and PPO in Figure 3 on Olympiad regarding two more efficiency aspects, i.e., training steps and wall-clock time. Our method demonstrates superior training efficiency by achieving a score above 40 on Olympiad with a 6x and 4x reduction in rollout data volume and wall-clock time. We provide more training curves for the other four benchmarks in Appendix K.

## 4.4 ANALYSIS ON THE EFFECTS OF OFF-POLICY RL FOR LLM RFT

In this subsection, we present an empirical analysis to gain better understanding of the effects of off-policy RL on LLM RFT. For convenience, we use ReMix-PPO for the analysis in the following.

To delve into the influence of off-policy RL enabled by ReMix on the reasoning behaviors during the learning process of LLMs, we make use of two more metrics: relative response length (against the training dynamics of PPO), and **self-reflection rate** that is calculated according to the occurrence of reflection tokens (e.g., 'verify', 'check', 'but', 'wait', etc.). Moreover, we compare PPO, ReMix-PPO, Mix-PPG and Mix-PPG with an increased UTD. The results are shown in Figure 4.

The vanilla PPO shows a steady increase of Pass@1 accuracy as well as a decrease in response length, while maintaining a self-reflection rate near 1. Mix-PPG accelerates early training but yields inferior asymptotic performance (see more in Figure 9), with a clear drop in response length and self-reflection rate. When applying an increased UTD ratio, Mix-PPG speeds further yet shows a destructive degradation after 200 steps, accompanied by a sharp decrease in response length and

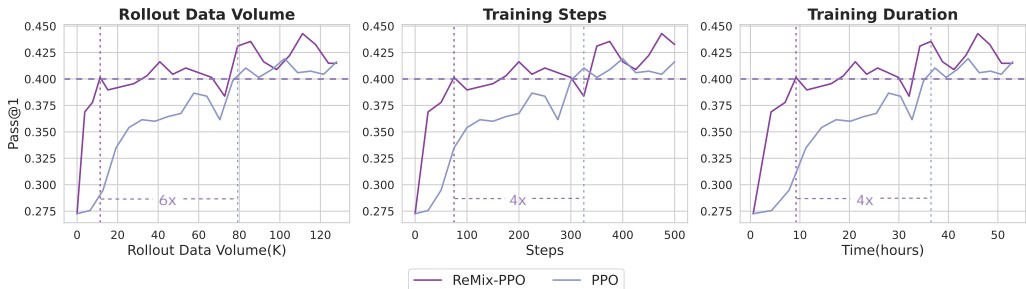

Figure 3: **Training efficiency comparison for ReMix-PPO and PPO (1.5B) on Olympiad.** We evaluate training efficiency across three dimensions: rollout data volume, training steps, and training duration. ReMix achieves a score above 40%, around **4× to 6× faster** than PPO.

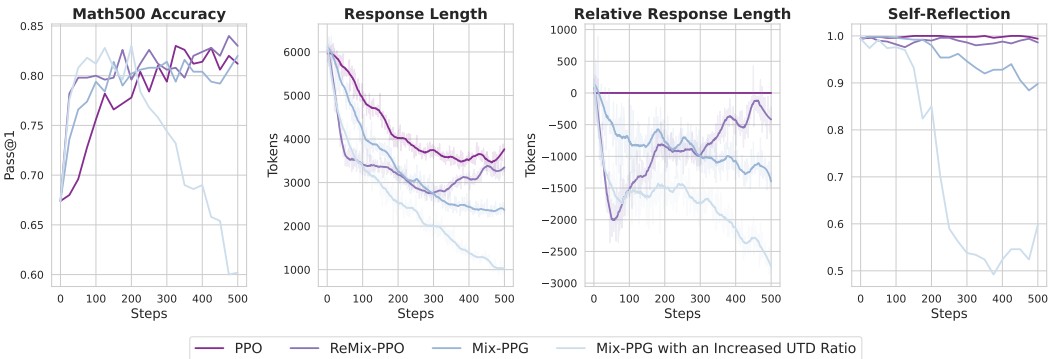

Figure 4: **Training dynamics regarding accuracy, response length, self-reflection rate for on-policy v.s. off-policy training.** ReMix shows a merged learning behavior and **perfectly combines the superior efficiency and the asymptotic improvement** thanks to the policy reincarnation.

self-reflection rate. This pathology drives the model to generate a final answer without adequate intermediate deliberation, resulting in a drop of accuracy. Please see Appendix M for detailed cases.

ReMix seems to perfectly combine the early-stage efficiency of Mix-PPG with an increased UTD and the asymptotic improvement of PPO, thanks to policy reincarnation. Throughout training, ReMix first quickly decreases the response length and improve its accuracy in the early stage; it then lengthens responses and uses more reflection for careful exploration and further improvement of accuracy.

> **Takeaway 3. A trade-off between efficiency and final performance needs to be balanced when incorporating off-policy training in RFT.**
>
> More off-policy training leads to a faster early-stage boost with a larger policy shift, leading to shorter responses and quicker unlearning of self-reflection, consequently damaging reasoning performance. ReMix well leverages off-policy efficiency at no compromise of final accuracy.

**The Implicit Preference of Off-policy Learning for Shorter Responses** To take a further step on why off-policy learning leads to the observed reasoning behaviors, we conduct a formal analysis on the learning dynamics when optimizing the Mix-PPG loss function $L_k^{\text{Mix-PPG}}(\theta)$ (shown in Eq. 3). Similarly as in (Fatemi et al., 2025), the average loss of Mix-PPG can be formulated below:

$$L_{\text{Avg}}^{\text{Mix-PPG}} = \frac{1}{H} \sum_{h=0}^{H} L_h^{\text{Mix-PPG}} \propto -\frac{1}{H} \sum_{h=0}^{H} r_\theta^{k-i} A_h^{\pi_k} \tag{6}$$

A simple derivation is provided in Appendix D.4. With the equation above, we can find: when the advantage estimate is *negative*, the model learns to minimize the loss by steering its policy to achieve a lower importance sampling ratio. As the policy loss is almost always positive (as in Figure 10), the advantage estimates are negative most of the time in our experiments. Also, the importance sampling ratio stays above one empirically (as in Figure 9), directly amplifies the loss term. Since the average loss is computed based on the data of historical policy $\pi_{k-i}$, there apparently exists a *Whipping Effect*:

the longer the response is, the larger the distribution shift should be on later states. Consequently, the model tends to prefer shorter responses to reduce the loss associated with long rollout trajectories. This tendency is further amplified as the proportion of off-policy data increases.

## 4.5 MORE RESULTS ON PASS@K METRICS, CODE GENERATION

**Pass@K Evaluation**  In the experiments above, we mainly use **Pass@1** accuracy (as well as Avg@32) as the evaluation metric. As advocated in (Yue et al., 2025), we extend our evaluation by using **Pass@8/16** accuracy here. The results are summarized in Table 4 and 5. We can observe that ReMix-PPO effectively improves both the Pass@8 and Pass@16 accuracies of the base model.

Table 4: **Pass@8 / Pass@16 accuracy (%) of 1.5B models on five reasoning benchmarks.**

| Model | AIME'24 | AMC'23 | MATH500 | Minerva | Olympiad | Avg. |
|---|---|---|---|---|---|---|
| R1-Distill-Qwen-1.5B (Base, Pass@8) | 20.00 | 44.58 | 73.00 | 19.12 | 31.70 | 37.68 |
| **ReMix-PPO** (Pass@8) | 30.00 | 68.67 | 84.60 | 28.68 | 46.67 | 51.72 |
| R1-Distill-Qwen-1.5B (Base, Pass@16) | 20.00 | 48.19 | 75.40 | 18.75 | 32.59 | 38.99 |
| **ReMix-PPO** (Pass@16) | 30.00 | 72.29 | 86.80 | 30.15 | 46.96 | 53.24 |

Table 5: **Pass@8 / Pass@16 accuracy (%) of 7B models on five reasoning benchmarks.**

| Model | AIME'24 | AMC'23 | MATH500 | Minerva | Olympiad | Avg. |
|---|---|---|---|---|---|---|
| R1-Distill-Qwen-7B (Base, Pass@8) | 63.33 | 83.13 | 93.80 | 47.79 | 55.11 | 68.63 |
| **ReMix-PPO** (Pass@8) | 76.67 | 92.77 | 95.80 | 50.74 | 63.70 | 75.94 |
| R1-Distill-Qwen-7B (Base, Pass@16) | 73.33 | 85.54 | 94.80 | 51.10 | 57.19 | 72.39 |
| **ReMix-PPO** (Pass@16) | 80.00 | 92.77 | 95.40 | 53.31 | 65.33 | 77.36 |

**ReMix for Code Generation**  Beyond Math reasoning, we move on to code generation to evaluate the domain generalization of ReMix. We use Skywork-OR1-RL-Data (He et al., 2025) for training. We set the max prompt length to 1600 because coding prompts are longer than math tasks. We evaluate our models on LiveCodeBench(8/1/24-2/1/25) (Jain et al., 2025) with sampling temperature of 0.7. All other experimental settings follow those in Section 4.1. The results are shown in Table 6. For both 1.5B and 7B scales, ReMix-PPO improves PPO while using much less cost, which aligns with our observation in the Math reasoning domain. This demonstrates the generalization ability of our method across both domains.

Table 6: **Performance evaluation of ReMix for code generation.** Pass@1 accuracy (%) on LiveCodeBench(8/1/24-2/1/25).

| Model | LiveCodeBench | Cost |
|---|---|---|
| R1-Distill-Qwen-1.5B (Base) | 12.09 | N/A |
| PPO (375 Steps) | 19.35 | 0.096M |
| **ReMix-PPO (300 Steps)** | 20.07 | 0.066M |
| **ReMix-PPO (475 Steps)** | 24.01 | 0.111M |
| R1-Distill-Qwen-7B (Base) | 26.16 | N/A |
| PPO (200 Steps) | 33.33 | 0.051M |
| **ReMix-PPO (100 Steps)** | 34.41 | 0.015M |

**Other Evaluation and Analysis**  We present supplementary performance evaluations to demonstrate the robustness of ReMix, encompassing experiments that use Dr.GRPO as a baseline in Appendix J.2 and Llama-series model as the base model in Appendix J.3. Additionally, we provide more analysis on the performance under response length constraint, the impact of prompt format, etc. Please refer to Appendix L for complete analysis results.

## 5 CONCLUSION

In this paper, we aim to address the notorious drawback of on-policy RFT methods (e.g., PPO and GRPO) on training inefficiency and prohibitive computational cost. We launch the renaissance of off-policy RL and propose Reincarnating Mix-policy Proximal Policy Gradient (ReMix), a general approach to enable on-policy RFT methods like PPO and GRPO to leverage off-policy data. In our experiments, we implement ReMix upon PPO, GRPO, and 1.5B-, 7B-scale base models. Through evaluating the reasoning accuracy and training efficiency of ReMix on five math reasoning benchmarks against 15 recent advanced baseline models, we demonstrate the superiority of ReMix in improving training efficiency and achieving SOTA-level reasoning performance with a great reduction in training cost. Due to the space constraint, we provide the discussion on limitations in Appendix C.

## ACKNOWLEDGEMENTS

This work is supported by the National Natural Science Foundation of China (Grant Nos. 62422605, 62533021, 62541610, 92370132, 62506186), the Fundamental Research Program of Shanxi Province (No. 202503021212091), the National Key Research and Development Program of China (Grant No. 2024YFE0210900), the Shanghai Municipal Science and Technology Major Project, and the Shanghai Artificial Intelligence Laboratory.

## ETHICS STATEMENT

We adhere to the code of ethics and the general principles. Our study fine-tunes publicly released LLM using reinforcement learning to improve mathematical reasoning, which involves no human subjects, user studies, or personal data. All datasets used for training and evaluation are publicly available. These resources contain no personally identifiable information or otherwise harmful information.

To mitigate potential risk, our release will be research-only. The authors declare no conflicts of interest and no sponsorship that would unduly influence the research.

## REPRODUCIBILITY STATEMENT

The code is provided in the following repository https://github.com/AnitaLeungxx/ReMix-Off-Policy-RFT. All experimental settings, hyperparameters and datasets are listed in Subsection 4.1 and Appendix I; descriptions of compared baselines can be seen in Appendix F. Any experiment-specific configurations are introduced in their respective sections. All datasets, base models and baselines we used are publicly available in HuggingFace.

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

CONTENTS OF APPENDIX

## A  THE USE OF LARGE LANGUAGE MODELS (LLMS)

We used LLMs only for coding and writing assistance. During experimentation, we consulted the LLM for code debugging. All algorithmic designs, implementations, and results were produced and verified by the authors, and LLM suggestions were reviewed and tested. After completing the manuscript, we used LLMS solely to polish the language (grammar and phrasing) without generating new scientific content. The authors remain fully responsible for the paper's contents.

## B  RELATED WORK

Post-training enhancement of LLM reasoning capabilities predominantly follows two paradigms (Li et al., 2025b). The first, inference-time optimization, improves reasoning without updating model parameters through techniques like Chain-of-Thought (CoT) prompting (Wei et al., 2022), parellel reasoning and itegration (Wang et al., 2022), self-reflection (Ji et al., 2023), tree-based search (Zhang et al., 2024), and macro-action-guided cognitive reasoning (Liu et al., 2025a). Despite their effectiveness, the performance of these methods is fundamentally constrained by the model's inherent capabilities. The second paradigm, parameter fine-tuning, aims to enhance these intrinsic abilities into LLM. While SFT on high-quality reasoning data is a common approach, its effectiveness is often limited by data availability and scalability (Zelikman et al., 2024). Consequently, RLVR has emerged as a powerful alternative, learning directly from reward signals to unlock superior performance, as demonstrated by models like DeepSeek-R1 (Guo et al., 2025). Notably, this differs from preference-based RL which learns from a reward model trained on human/AI feedback (Bai et al., 2022a;b; Liu et al., 2024), as the RLVR here utilizes direct, verifiable reward signals. Our work is situated within the RFT paradigm, especially under varifirable reward.

The majority of existing RFT research has relied on on-policy RL algorithms prized for their training stability, such as PPO (Schulman et al., 2017). Some recent approaches have sought to improve efficiency by modifying the RL architecture (e.g., GRPO (Shao et al., 2024)) or relaxing optimization constraints (Seed et al., 2025). However, these on-policy RL methods exhibit severe sample inefficiency, as they require fresh samples for each iteration of gradient updates. To alleviate this, recent research has begun to incorporate off-policy data in RL training. Tang et al. (2025) propose AGRO for a unified algorithm to leverage any-generation data, encompassing both on- and off-policy samples. However, their experimental results show that off-policy training is inferior to on-policy training, underscoring the non-trivial challenge of achieving stable and effective off-policy training for LLMs. Tapered Off-Policy REINFORCE (Roux et al., 2025) introduces a novel variant of importance sampling to downweight negative trajectories that are not likely under the current policy, while allowing positive trajectories to be upweighted. This enables the utilization of both off-policy and on-policy rollout trajectories. The method is trained and evaluated on GSM8K and MATH, leaving its efficacy on broader reasoning tasks unknown.

Recently, concurrent to our work, Based on REINFORCE, AsymRE (Arnal et al., 2025) is proposed to leverage both off-policy and on-policy data by introducing a tunable baseline. An asymmetry is presented that while on-policy updates safely leverage both positive and negative signals, off-policy updates benefit more from positive rewards, which to some extent echoes the idea proposed in (Roux et al., 2025). AsymRE is trained and evaluated on MATH. SRPO (Zhang et al., 2025b) builds on GRPO with a two-stage curriculum that first trains on math and then on code. Besides, SRPO adopts historical resampling discards groups with uniform rewards to avoid zero gradients and retains hard samples for later replay. Using the same base model, SRPO outperforms DeepSeek-R1-Zero-Qwen-32B while using only one-tenth of the training steps. Similarly, RePO (Li et al., 2025a) also exploits historical data. SRPO's resampling chiefly targets the quality of samples, while RePO emphasizes efficiency and systematically analyzes the impact of replay strategies. RePO is proposed upon GRPO to replay both historical off-policy data and on-policy data together during typical GRPO training. Different off-policy data replay strategies are studied, among which recency-based and reward-based strategies show improved performance. The RePO models are trained with a maximum response length of 1,024, thus showing limited performance on math reasoning benchmarks.

By following the principle of Soft RL, SPO (Cohen et al., 2025)) is proposed to leverage both off-policy and on-policy data based on Cumulative Q-Parameterization. SPO is trained and evaluated for code contests and demonstrates superior performance to the standard PPO. In contrast, LUFFY (Yan

et al., 2025) uses off-policy samples from superior models (e.g., DeepSeek-R1) and employing policy shaping. However, in essence, this is more akin to learning from demonstrations rather than the canonical off-policy RL where the behavior policy is often one of the historical policies or a separate inferior policy. Moreover, the idea of off-policy guidance is orthogonal to our method.

While early efforts have conducted first-step explorations on realizing off-policy learning for RFT, they have primarily focused on adapting existing on-policy methods (e.g., PPO, GRPO, REINFORCE) to off-policy data from the angles of modifying importance sampling, leveraging data or trajectories asymmetrically, etc. These initial steps have not investigated the essential effects of off-policy learning on reasoning behaviors, while leaving the potential of existing off-policy RL techniques unexplored. In the broader field of RL, methods such as Rainbow (Hessel et al., 2018), TD3 (Fujimoto et al., 2018), and SAC (Haarnoja et al., 2018) have set a precedent for leveraging historical data to improve sample efficiency. Building on this, advanced research has pursued maximizing data utilization through high UTD ratios, managing the resultant estimation errors with techniques like ensemble learning, as seen in REDQ (Chen et al., 2021), DroQ (Hiraoka et al., 2021), and SPEQ (Romeo et al., 2021). Concurrently, novel approaches have emerged, including hybrid methods that seek an optimal balance between the stability of on-policy learning and the efficiency of off-policy methods (Queeney et al., 2021), as well as fully offline algorithms designed to mitigate extrapolation errors from static datasets (Ma et al., 2024b; 2023). The value of ReMix lies in its departure from simply implementing off-policy RL in the context of RFT. Instead, by drawing inspiration from rich RL literature, our research aims to conduct an in-depth investigation of different off-policy RL techniques and integrate them to improve the RFT process effectively, thereby significantly enhancing the efficiency and performance of LLM fine-tuning.

## C    LIMITATIONS

Due to resource constraints, our experiments were limited to models up to 7B. While this provides a strong proof-of-concept, performance on larger-scale models is yet to be explored. To support this future work, we provide open-source code and models for community validation. We also note that the principle of using off-policy data to improve sample efficiency is general and not inherently tied to model scale. For the utilization of off-policy data, we use fixed proportions in this work, although we believe an adaptive control on the proportion of off-policy data should be possible and favorable. Moreover, our method is orthogonal to many of the advanced RFT methods considered and not considered in our experiments, while we do not explore the combination of them. We believe that integrating off-policy learning and other advanced techniques is promising to realize new LLM models that are more efficient and powerful at the same time. We leave these potential angles for the future.

## D    ALGORITHM AND RELATED DISCUSSION

### D.1    THE PSEUDOCODE OF REMIX

The pseudocode of ReMix is presented in Algorithm 1.

### D.2    THEORETICAL DISCUSSION ON THE HISTORICAL POLICY DISTRIBUTION $\nu$

The distribution of policy index $i$, where $i \in \{0, 1, \ldots, N\}$, is denoted by $\nu$. The theoretical explanation for the influence of different choices of $\nu$ should trace back to how Trust-Region Policy Optimization (TRPO) (Schulman et al., 2015) approximates the trust region.

For $i = 0$, i.e., the standard PPO (on-policy case), the trust region $\alpha^2 = [\max_s TV(\pi_k, \pi)(s)]^2$ or its upper bound $D_{\mathrm{KL}}(\pi_k, \pi)$ (this is can be found in Theorem 1, Eq.8 in TRPO paper) is approximated by using the expectation regarding $d^{\pi_k}$ to replace the maximum case. In turn, the approximate trust region is $\tilde{\alpha}^2 = [\mathbb{E}_{s \sim d^{\pi_k}} TV(\pi_k, \pi)(s)]^2$. In the practices of PPO and TRPO, this approximation works well in many problems. Intuitively, this is because the new policy $\pi$ (post-update) should not differ a lot with the current policy $\pi_k$ (prior-update), the distribution $d^{\pi_k}$ works as an effective surrogate.

---

**Algorithm 1** Reincarnating Mix-Policy Proximal Policy Gradient Method (**ReMix**)

---

1: [Input]: Base model $\pi_{\text{base}}$, and on-policy proximal PG method $\mathbb{A}$ (e.g., PPO, GRPO)
2: Set training batch size $B$, off-policy data portion $p$, UTD ratio $m$, historical policy window size $N$, policy reincarnation step point $T$
3: Init the model $\pi_\theta = \pi_{\text{base}}$ and the historical policy set $\mathbb{H} = \emptyset$
4: # Stage 1: Mix-policy Proximal PG Training
5: **for** step $t = 1, 2, 3, ..., T$ **do**
6:     Sample a batch of questions $q \sim \mathcal{D}_0$ with size $(1 - p)B$ and generate fresh responses according to $\pi_\theta$ and $\mathbb{A}$
7:     Reuse historical responses from $\mathbb{H}$ with size $pB$ and form the mixed training batch
8:     Save $\pi_\theta$ to $\mathbb{H}$ with its responses and logprob data, drop the oldest policy if $|\mathbb{H}| > N$
9:     Repeatedly update $\pi_\theta$ with the mixed training batch according to Mix-PPG and $\mathbb{A}$ (the first row, Eq. 5) for $m$ times
10: **end for**
11: # Stage Transition: Policy Reincarnation
12: Reset the base reference model from $\pi_{\text{base}}$ to $\pi_T$, and drop the historical policy set $\mathbb{H}$
13: # Stage 2: Reincarnating On-policy Proximal PG Training
14: **for** step $t = T + 1, T + 2, T + 3, ...$ **do**
15:     Sample a batch of questions $q \sim \mathcal{D}_0$ with size $B$ and generate responses according to $\pi_\theta$ and $\mathbb{A}$
16:     Construct a training batch with the fresh responses, and update $\pi_\theta$ according to $\mathbb{A}$ (the second row, Eq. 5)
17: **end for**

---

When it moves on to the off-policy case where $i > 0$ (Queeney et al., 2021), the approximation for the trust region could be no longer effective for stable policy optimization due to the increasing discrepancy between $d^\pi$ and $d^{\pi_{k-i}}$ as the increase of $i$ (i.e., for older historical policies). In our work, we empirically found that using too old historical policies can introduce large off-policyness which makes the training unstable.

Our empirical observation aligns with our discussion on the theoretical explanation of the approximation of trust region above. Therefore, in practice, we use $N = 2$, $p = 0.4$ and a uniform distribution for off-policy data. This is equivalent to the policy index distribution $\nu = [0.6, 0.2, 0.2]$. We believe that one important future direction is to study how to replay off-policy data better instead of replaying in a uniform manner.

### D.3 THEORETICAL DISCUSSION ON THE KL-CONVEX LOSS

The KL-convex loss is theoretically grounded in Conservative Policy Iteration (CPI) (Ma et al., 2024a). The CPI paper proves that iteratively refining the reference policy guarantees monotonic improvement (Proposition 1: $V^{\bar{\pi}^*}(s) \geq V^{\bar{\pi}}(s)$) and support preservation ($\bar{\pi}^*(a|s) = 0$ wherever $\bar{\pi}(a|s) = 0$). Our convex combination $L_{\text{KLC}} = \lambda D_{\text{KL}}(\pi_\theta || \pi_{\text{base}}) + (1 - \lambda) D_{\text{KL}}(\pi_\theta || \pi_{k-1})$ directly implements this — where the $\pi_{k-1}$ term acts as CPI's dynamic reference policy to prevent OOD queries, while the $\pi_{\text{base}}$ term extends the framework to preserve foundational capabilities.

Furthermore, CPI's Theorem 1 identifies that multi-step actor-critic implementations suffer from high variance, and the authors explicitly recommend adding behavior regularization (their Eq. 6) to constrain policies to data support. Our decaying $\lambda(t)$ schedule dynamically balances this trade-off, shifting from conservative exploration to aggressive refinement. Thus, KLC inherits CPI's theoretical guarantees of improvement and convergence while addressing practical stability challenges in iterative LLM fine-tuning.

### D.4 THE DERIVATION OF EQUATION 6

We conduct the formal analysis of off-policy PPG in Section 4.4 by using a similar form of average loss as in (Fatemi et al., 2025). Here, we provide the complete derivation of Equation 6 below.

Starting from Equation 3, that is the definition of the loss function for Mix-PPG, we rewrite the loss function from the original per-sample expectation (i.e., $(s, a) \sim d^{\pi_{k-i}}$) form to the per-trajectory

expectation form (i.e., $\tau \sim d^{\pi_{k-i}}$). This derives:

$$
L_{\text{Avg}}^{\text{Mix-PPG}}(\theta) = -\mathbb{E}_{i \sim \nu}\Big[\mathbb{E}_{\tau \sim d^{\pi_{k-i}}} \frac{1}{H_\tau} \Sigma_{h=0}^{H_\tau} \min\Big(r_\theta^{k-i}(s_h, a_h) A^{\pi_k}(s_h, a_h),
$$
$$
\text{clip}\Big(r_\theta^{k-i}(s_h, a_h), \frac{\pi_k(a_h \mid s_h)}{\pi_{k-i}(a_h \mid s_h)} - \epsilon, \frac{\pi_k(a_h \mid s_h)}{\pi_{k-i}(a_h \mid s_h)} + \epsilon\Big) A^{\pi_k}(s_h, a_h)\Big)\Big], \tag{7}
$$

where $H_\tau$ is response length for trajectory $\tau$. By dropping the clipping range (i.e., ignoring the out-of-clipping range cases which have no gradient) and simplifying the expression by omitting subscript notations, expectation notations, etc., we only keep the proportional relationship for the analysis. This then leads to: $L_{\text{Avg}}^{\text{Mix-PPG}} = -\frac{1}{H}\Sigma_{h=0}^{H} r_\theta^{k-i} A^{\pi_k}$, which is the exact form used in Equation 6.

### D.5 Sensitivity Analysis of The Choices of Policy Reincarnation Trigger Step $T$

In our pipeline, $T$ is the hand-over step that switches from the off-policy Mix-PPG phase to the on-policy phase with reference model changed too. We use it to harvest early data-efficiency from replay historical data and then let on-policy optimization continue improving stably. To address the reviewer's concern about how the trigger step $T$ was chosen and how sensitive performance is to this choice, we provide a controlled sweep $T \in \{25, 50, 100, 200\}$ with 500-step runs, whose experimental settings are all the same as those of the main text. The results are shown in the table below. For ($T = 25$), we report two checkpoints per run, the first time the macro average reaches $\geq 49$ ("early lift") and the final plateau

The pattern is consistent: a very small $T$ (25) slows down performance improvement; a very large $T$ (200) lifts early but later stalls due to amplified off-policyness; a moderate $T$ (50–100) yields both a strong early lift and the best final averages. We also find this choice to broadly work well across our experimental cases, achieving a favorable efficiency-performance trade-off

Table 7: **Pass@1 accuracy (%) of 1.5 B model under different policy reincarnation trigger step $T$.**

| Model | AIME'24 | AMC'23 | MATH500 | Minerva | Olympiad | Avg. |
|---|---|---|---|---|---|---|
| $T = 25$ (at 350 steps) | 33.33 | 65.06 | 81.20 | 27.57 | 39.26 | 49.29 |
| $T = 50$ (at 75 steps) | 43.33 | 63.86 | 79.60 | 26.84 | 39.41 | 50.61 |
| $T = 50$ (at 500 steps) | 40.00 | 63.86 | 83.00 | 26.84 | 43.70 | 51.52 |
| $T = 100$ (at 225 steps) | 30.00 | 69.88 | 81.80 | 24.63 | 41.78 | 49.62 |
| $T = 100$ **(at 325 steps)** | **36.67** | **69.88** | **82.00** | **30.15** | **41.78** | **52.09** |
| $T = 200$ (at 100 steps) | 30.00 | 66.27 | 81.00 | 29.41 | 43.26 | 49.99 |
| $T = 200$ (at 500 steps) | 16.67 | 49.40 | 77.00 | 16.67 | 38.67 | 40.76 |

## E Advantage Estimation

To enable stable off-policy training, we adopt a V-trace (Espeholt et al., 2018) formulation for generalized advantage estimation (GAE) (Schulman et al., 2016), which incorporates truncated importance sampling ratios to correct for policy mismatch. We first compute the temporal-difference error(TD-error) at each time step $t$ as

$$
\delta_t^V = r(s_t, a_t) + \gamma V(s_{t+1}) - V(s_t), \tag{8}
$$

and define the truncated importance sampling weight $c_t = \min\left(\bar{c}, \frac{\pi_k(a|s)}{\pi_{k-i}(a|s)}\right)$, where $\bar{c}$ is a clipping threshold to limit the variance of the correction, we use $\bar{c} = 1$ in our implement.

The advantage at step $t$ is estimated recursively using the V-trace correction as

$$
A_t = \delta_t^V + \gamma\lambda c_t A_{t+1}, \tag{9}
$$

and the return-to-go is computed by combining the advantage estimate with the baseline value:

$$
\text{RTG}_t = A_t \cdot c_t + V(s_t). \tag{10}
$$

This V-trace corrected GAE formulation ensures that the estimated advantages remain stable and consistent under significant off-policy drift, which is critical in our training regime involving long-horizon trajectories and evolving policies.

## F  A Brief Overview of Baseline Models

### F.1  1.5B Models

- **Open-RS Series** (Dang & Ngo, 2025): The Open-RS series employs the GRPO algorithm to train language models, using datasets constructed by filtering and combining existing corpora. Specifically, **Open-RS1** utilizes dataset with 18,615 samples with accuracy and format rewards, **Open-RS2** incorporates dataset with 7,000 samples and shorter maximum response length while retaining the same reward functions. Compared to **Open-RS2**, **Open-RS3** replaces the accuracy reward with a cosine reward and adds an English-only instruction to the system prompt.

- **DeepScaleR** (Luo et al., 2025): DeepScaleR is obtained via a two-phase training process with the GRPO algorithm: starting with 8k context for efficient reasoning, then scaling up to 16k and 24k contexts to address more challenging problems.

- **II-Thought** (Intelligent-Internet, 2025): Based on a systematic analysis of existing public datasets, the authors constructed a large-scale, high-quality dataset comprising over 300,000 reasoning problems across multiple domains. Each sample was rigorously filtered and deduplicated. Subsequently, the models were trained on this curated dataset, using the GRPO algorithm.

- **FastCuRL Series** (Song et al., 2025): The FastCuRL Series adopts a multi-stage training process where both context length and data complexity (defined by input prompt length) are progressively increased. Training starts with short-context and low-complexity data, then moves to longer contexts with medium and high-complexity datasets.

- **L1 Series** (Aggarwal & Welleck, 2025): The L1 Series trains models using Length-Controlled Policy Optimization (LCPO), a method that encourages correct answers while matching a target output length specified in the prompt (measured by input prompt length). **L1-Exact** enforces exact-length generation by penalizing deviation from the target length, while **L1-Max** applies a soft maximum-length constraint, allowing shorter outputs when appropriate but discouraging overruns.

- **AdaptThink** (Zhang et al., 2025a): AdaptThink is an RFT method that trains reasoning models to choose between two modes — Thinking and NoThinking — based on problem difficulty. It uses a constrained optimization objective to encourage NoThinking while maintaining performance, and an importance sampling strategy to balance both modes during training.

### F.2  7B Models

- **ReasonFlux-F1** (Yang et al., 2025b): ReasonFlux-F1 is an SFT model obtained by fine-tuning an R1-Distill model[3] based on template-augmented reasoning trajectories collected by ReasonFlux-v1. These trajectories are first enhanced with structured templates, then transformed into a long chain-of-thought format.

- **Light-R1** (Wen et al., 2025): Light-R1 is a multi-stage post-training framework. It begins with curriculum-based supervised fine-tuning (SFT) using progressively harder data, followed by Direct Preference Optimization (DPO) and an RFT process with GRPO on a filtered dataset. `Light-R1-7B-DS` is trained only in the second SFT stage of the framework. Thus, the Light-R1 7B baseline model used in our experiments is an SFT model rather than an RFT model.

- **Skywork-OR1-Preview** (He et al., 2025): Skywork-OR1-Preview is trained on a curated dataset of math and coding problems, selected through model-aware difficulty estimation and quality filtering. The training process modifies GRPO by incorporating both offline and online difficulty-based filtering, rejection sampling, and a multi-stage curriculum with adaptive entropy control.

- **Polaris** (An et al., 2025): Polaris adopts a multi-stage RL training approach with careful data difficulty control, using a data distribution with a slight bias toward challenging problems and

---

[3] https://github.com/Gen-Verse/ReasonFlux/blob/main/ReasonFlux_F1/README.md

dynamically adjusting question difficulty during training. It initializes sampling temperature based on rollout diversity and gradually increases it during training. It employs length extrapolation techniques, enabling longer CoT generation at inference while keeping training rollouts short.

- **AdaptThink** (Zhang et al., 2025a): The methodology for the AdaptThink 7B model is identical to that of the AdaptThink 1.5B model, as previously described.

- **AceReason-Nemotron** (Chen et al., 2025): AceReason-Nemotron adopts the GRPO algorithm without KL divergence and avoids entropy collapse through controlled updates. The model is first trained on math-only prompts, then on code-only prompts, following a curriculum with progressively increasing response lengths.

### F.3    EXCLUSION RATIONALE FOR OFF-POLICY BASELINES

For existing off-policy RFT methods, we do not include RePO (Li et al., 2025a) because their models are trained under a maximum response length of 1,024 tokens, thus showing limited performance on math reasoning tasks. We do not include LUFFY (Yan et al., 2025) since the usage of off-policy guidance from a superior model (e.g., DeepSeek-R1) is orthogonal to ReMix, which is also viewed as a different setting where extrinsic guidance or demonstrations are accessible. We exclude SRPO (Zhang et al., 2025b) since its publicly released model is not at the same scale as ours. In addition, we did not find public checkpoints for SPO (Cohen et al., 2025) (which is also trained for code contests), AGRO (Tang et al., 2025), AsymRE (Arnal et al., 2025) and Tapered Off-policy REINFORCE (Roux et al., 2025), thus, we do not include them in our experiments. Please refer to Section B for detailed discussions on related off-policy RFT methods.

## G    SYSTEM PROMPT

Following the standard DeepScaler data processing approach, each prompt in the training set was prefixed with "<|User|>" and suffixed with the instruction "Let's think step by step and output the final answer within \boxed{}. <|Assistant|><think>" . This structure encourages the model to engage in step-by-step reasoning and produce final answers encapsulated within LaTeX boxed expressions. One example of the DeepScaler prompt format is shown below. The blue text indicates the fixed template used during inference, while the black text represents the instance-specific question inserted into the prompt.

---

**System Prompt (Standard)**

<|begin_of_sentence|><|User|> Xenia and Sergey play the following game. Xenia thinks of a positive integer $N$ not exceeding 5000. Then she fixes 20 distinct positive integers $a_1, a_2, \ldots, a_{20}$ such that, for each $k = 1, 2, \ldots, 20$, the numbers $N$ and $a_k$ are congruent modulo $k$. By a move, Sergey tells Xenia a set $S$ of positive integers not exceeding 20, and she tells him back the set $\{a_k : k \in S\}$ without spelling out which number corresponds to which index. How many moves does Sergey need to determine for sure the number Xenia thought of? Let's think step by step and output the final answer within \boxed{}. <|Assistant|><think>

---

## H    KEY OBSERVATIONS FROM FIGURE 1: EFFICIENCY−ACCURACY

For clarity, we summarize the major observations in Figure 1 below:

- **(1.5B) ReMix-PPO v.s., DeepScaleR:** DeepScaleR, the strongest 1.5B competitor, requires around **2.519M** rollouts to reach its final score (i.e., **52.14**), whereas ReMix-PPO (350 Steps) achieves a comparable score (i.e., **52.10**) with **0.079M** rollouts — over **a 30x reduction** in rollout data volume.

- **(1.5B) ReMix-PPO v.s., PPO:** We trace the performance of ReMix-PPO at 100, 200, and 350 training steps (denoted by the yellow curve in Figure 1), corresponding to rollout data volumes of roughly 0.020M, 0.041M, and 0.079M, respectively. Even after generating just

**0.020M** rollout samples, ReMix-PPO achieves a score of **50.61**, which has already surpasses most baselines. Compared to PPO (900 Steps), which achieves an average score of **50.61** with **0.230M** rollouts, our model shows over **a 10x reduction** in rollout data volume.

- **(1.5B) ReMix-GRPO v.s., GRPO:** We also trace the performance of ReMix-GRPO at 50, 100, and 200 training steps (denoted by the cyan curve in Figure 1), corresponding to rollout data volumes of roughly 0.061M, 0.163M, and 0.368M, respectively. After generating **0.061M** rollout samples, our model achieves the score **45.47** that exceeds the score **45.20** of standard GRPO trained for 100 steps with **0.205M** rollout samples. Compared to GRPO (200 Steps), which achieves an average score of **48.64** with **0.410M** rollouts, ReMix-GRPO achieves a much higher score of **50.53** within 200 training steps, i.e., **0.368M** rollouts, showing a superior final performance with less computational cost.

- **(7B) ReMix-PPO v.s., AceReason-Nemotron:** AceReason-Nemotron, the strongest 7B baseline method in our comparison, requires over **3.584M** rollouts to reach its final score (i.e., **63.24**)[4], whereas ReMix-PPO (50, 75 Steps) achieves a slightly higher accuracy (i.e., **63.27**, **64.39**) with **0.007M**, **0.011M** rollouts — over **a 450x reduction** in rollout data volume.

- **(7B) ReMix-PPO v.s., AdaptThink:** AdaptThink, the second strongest 7B baseline method, requires around **0.307M** rollouts to reach its final score (i.e., **58.77**), whereas ReMix-PPO (25 Steps) achieves a comparable accuracy (i.e., **58.49**) with **0.003M** rollouts — over **an 80x reduction** in rollout data volume.

- **(7B) ReMix-PPO v.s., PPO:** Compared to PPO (200 Steps) that achieves an average score of **60.34** with **0.051M** rollouts, ReMix-PPO achieves a higher score of **63.27** within 50 training steps, i.e., **0.007M** rollouts, showing **a 6x reduction** in rollout data volume.

Besides, when `do_sample` is set to `true`, Open-RS series models (i.e., -RS1, -RS2, -RS3) show better scores 40.62, 40.08, 39.31 respectively, and `II-Thought` can achieve a score 51.474. For other models, we found similar scores in our experiments, which do not change the conclusions.

## I  TRAINING DETAILS

**Hyperparmeters**  The major hyperparameter choices are shown in Table 8.

**Compute Resource**  The 1.5B model was trained for 50 hours on 2 NVIDIA A800-SXM4-80GB GPUs, while the 7B model required 75 hours on 8 such GPUs. The evaluation of each model was also conducted using the same number of GPUs as in their respective training setups.

**Comparison of Training Detail on Computational Cost for 1.5B Models**  The corresponding detailed factors associated with computational cost for training the 1.5B models in the comparison above are shown in Table 9. Compared to most baselines, our method uses nearly half the number of training steps (500 v.s. $\geq$ 860) while delivering superior performance. Furthermore, our entire training run is executed on a single node with just two A800 GPUs over 52 hours, amounting to 104 A800 GPU hours. This finding shows that state-of-the-art gains can be achieved with markedly reduced compute requirements.

**Comparison of Training Detail on Computational Cost for 7B Models**  Table 10 shows the training details of 7B models. However, we failed to find complete training details for all the 7B models, so we did not plot the efficiency-performance trade-off for the 7B models due to missing information.

## J  SUPPLEMENTARY PERFORMANCE EVALUATION

### J.1  EXTENDED EVALUATION OF AIME'24 AND AMC'23

For two small and high-variance benchmarks (i.e. AIME'24 and AMC'23), we report Pass@1 as **the average over 32 independent runs (Avg@32)**. Here we use use stochastic decoding (temperature

---

[4]The score of AceReason-Nemotron is obtained by evaluating the official checkpoint, and the rollout data volume is estimated according to the text and Figure 3 in (Chen et al., 2025).

Table 8: **Hyperparameter setups for PPO, GRPO and ReMix trainer.**

| Parameter | Value |
|---|---|
| *Training Configuration* | |
| temperature | 1.0 |
| top-p | 1.0 |
| top-k | -1 |
| critic_warmup | 0 |
| learning_rate | 1e-6 |
| clip_ratio | 0.2 |
| lam | 1 |
| tau | 0.95 |
| entropy_coeff | 0.001 |
| clipping_gradient | true |
| do_sample | true |
| test_freq | 25 |
| *Training Configuration for ReMix-GRPO and GRPO* | |
| kl_loss_coef | 0.001 |
| kl_loss_type | low_var_kl |
| n (gen per prompt) | 8 |

Table 9: **RFT training details associated with computational cost for 1.5B models.** All the models are trained upon `DeepSeek-R1-distill-Qwen2.5` base model, except for L1 series models, which are fine-tuned on top of `DeepScaleR` (denoted by superscript *). Accordingly, their total training cost should be considered as the sum of `DeepScaleR`'s cost and the resources reported in this table. *Italicized entries* indicate values not directly reported in the original papers, but instead retrieved from associated official training scripts. The underlined values denote the fresh on-policy rollout in addition to off-policy data reuse in ReMix.

| Model | Training Steps | Rollout Batch Size | Gen per Prompt | Max Responses Length | Number of GPUs |
|---|---|---|---|---|---|
| DeepScaleR | 1750 steps | *128,128,128* | *8,16,16* | 8k,16k,24k | 8,32,32 |
| FASTCuRL-preview | 860 steps | 128,64,64,64 | 8,8,8,16 | 8k,16k,24k,16k | 8 |
| FASTCuRL-v3 | 2620 steps | *128,64,64,64,64* | *8,8,8,16,16* | 8k,16k,24k,16k,16k | 8 |
| II-Thought | - | *1024* | *5* | 32k | *8* |
| adapt think | 314 steps | 128 | 16 | 16k | 8 |
| Open-RS1 | 100 steps | 96 | 6 | 4k | 4 |
| Open-RS2 | 50 steps | 96 | 6 | 4k | 4 |
| Open-RS3 | 50 steps | 96 | 6 | 4k | 4 |
| L1-Exact* | 700 steps | 128 | 16 | 4k | *8* |
| L1-Max* | 120 steps | 128 | 16 | 4k | *8* |
| **ReMix-PPO** | 500 steps | 152,256 | 1 | 8k | 2 |
| **ReMix-GRPO** | 200 steps | 152,256 | 8 | 8k | 2 |

Table 10: **RFT training details associated with computational cost for 7B models.** All methods are trained upon `DeepSeek-R1-Distill-Qwen-7B` base model. *Italicized entries* indicate values not directly reported in the original papers, but instead retrieved from associated official training scripts. The underlined values denote the fresh on-policy rollout in addition to off-policy data reuse in ReMix. Note that ReasonFlux-F1 and Light-R1 (7B) are SFT models as detailed in Appendix F.2, hence we do not include them in this table.

| Model | Traing Steps | Rollout Batch Size | Gen per Prompt | Maximum Responses Length | Number of GPUs |
|---|---|---|---|---|---|
| Skywork-OR1-Preview | >2000 steps | *256* | *16* | 8k,16k,32k | 8 |
| AceReason-Nemotron | >2000 steps | 128 | 8,16,16,16 | 8k,16k,24k,32k | 128 |
| AdaptThink | 150 steps | 128 | 16 | 16k | 8 |
| Polaris | >1400 steps | - | - | 16k,24k,32k | - |
| **ReMix-PPO** | 500 steps | 152,256 | 1 | 8k | 8 |

0.1) for evaluation. We include these results in Table 11 and Table 12 to provide a robust estimate on small datasets. ReMix-PPO maintains the similar performance observed in Table 1 and Table 2 under greedy decoding, showing consistent gains under the Avg@32 protocol.

Table 11: **Pass@1 accuracy (%) on AIME'24 and AMC'23 (Avg@32) of 1.5B models.** ReMix-PPO shows performance consistent with the greedy-decoding results (Table 1). **Bolded** and underlined values denote the highest and the second-highest scores in each dataset (i.e., column).

| Model | AIME'24 | AMC'23 |
|---|---|---|
| R1-Distill-Qwen-1.5B (Base) | 17.92 | 44.58 |
| Open-RS1 | 17.60 | 45.44 |
| Open-RS2 | 17.62 | 44.92 |
| Open-RS3 | 18.65 | 43.98 |
| AdaptThink | 19.06 | 56.81 |
| II-Thought | 28.65 | 59.72 |
| FASTCuRL-preview | 25.94 | 54.27 |
| FASTCuRL-V3 | **33.44** | 63.63 |
| L1-Exact* | 25.1 | 66.57 |
| L1-Max* | 23.13 | **66.79** |
| DeepScaleR | 31.96 | 63.58 |
| **ReMix-GRPO (75 Steps)** | 25.10 | 60.17 |
| **ReMix-PPO (350 Steps)** | 29.08 | 64.04 |

Table 12: **Pass@1 accuracy (%) on AIME'24 and AMC'23 (Avg@32) of 7B models.** ReMix-PPO reach SOTA-level on both benchmarks.

| Model | AIME'24 | AMC'23 |
|---|---|---|
| R1-Distill-Qwen-7B (Base) | 37.53 | 66.55 |
| ReasonFlux-F1 | 20.19 | 53.07 |
| Light-R1 | 40.00 | 66.73 |
| Skywork-OR1-Preview | 36.31 | 61.60 |
| Polaris | 39.71 | 67.40 |
| AdaptThink | 47.62 | 74.20 |
| AceReason-Nemotron | 50.00 | 77.48 |
| **ReMix-GRPO (200 Steps)** | 49.48 | 82.12 |
| **ReMix-PPO (75 Steps)** | **50.31** | 77.78 |

## J.2 REMIX FOR DR.GRPO

To further examine the generalization of ReMix regarding different base algorithms, we evaluate the effect of ReMix based on Dr.GRPO (Liu et al., 2025b). For the convenience of experimentation, we use `DeepSeek-R1-Distill-Qwen-1.5B` as the base model and set the max response length to 4096. We evaluate the models with sampling temperature of 0.7. All other experimental settings follow those in Section 4.1. The results show that ReMix-Dr.GRPO delivers consistent improvements over Dr.GRPO with higher efficiency, mirroring the finding in our main evaluation above. This demonstrates the generality of ReMix as an effective drop-in method.

Table 13: **Performance evaluation of ReMix based on Dr.GRPO.** Pass@1 accuracy (%) on five benchmarks (Avg@32 for AIME'24/AMC'23). The max response length is 4096.

| Model | AIME'24 | AMC'23 | MATH500 | Minerva | Olympiad | Avg. | Cost |
|---|---|---|---|---|---|---|---|
| R1-Distill-Qwen-1.5B (Base Model) | 9.38 | 32.42 | 63.40 | 18.75 | 22.67 | 29.32 | N/A |
| Dr.GRPO (350 steps) | 21.25 | 63.10 | 79.40 | 27.21 | 41.33 | 46.48 | 0.358 M |
| Dr.GRPO (400 steps) | 26.25 | 59.94 | 80.60 | 26.57 | 41.03 | 46.86 | 0.409 M |
| **ReMix-Dr.GRPO (325 steps)** | 26.25 | **63.25** | 81.60 | 27.21 | 41.63 | 47.09 | 0.291 M |
| **ReMix-Dr.GRPO (400 steps)** | **28.75** | 62.65 | **82.00** | **27.74** | **42.96** | **48.42** | 0.368 M |

## J.3 ReMix for Llama-series Base Model

In addition to the Qwen-series base model used above, we also evaluate the effect of ReMix based on `DeepSeek-R1-Distill-Llama-8B` in Table 14 in the appendix. Similarly, we found that ReMix-PPO improves the performance of the base model and outperforms PPO in terms of both score and efficiency.

Table 14: **Pass@1 accuracy (%) of R1-Distill-Llama-8B.**

| Model | AIME'24 | AMC'23 | MATH500 | Minerva | Olympiad | Avg. | Cost |
|---|---|---|---|---|---|---|---|
| R1-Distill-Llama-8B (Base Model) | 26.46 | 62.69 | 82.00 | 25.37 | 42.67 | 47.84 | N/A |
| PPO (50 steps) | 34.58 | 70.63 | 85.40 | 26.10 | 45.18 | 52.38 | 0.013 M |
| **ReMix-PPO (25 steps)** | **39.48** | **88.00** | **87.04** | **28.68** | **48.74** | **55.99** | **0.004 M** |

## K More Training Curves

**Training Curves for Efficiency Comparison** In addition to the efficiency comparison between ReMix-PPO and PPO for Olympiad in Figure 3, the remaining curves for the other four math reasoning benchmarks are presented in Figure 5, 6, 7, 8.

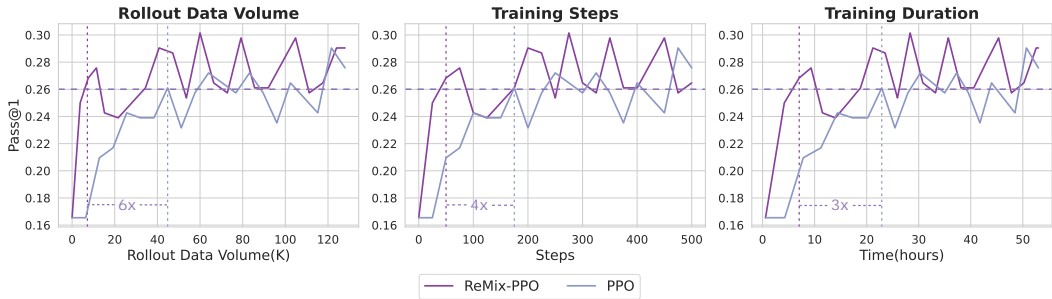

Figure 5: **Training efficiency comparison for ReMix-PPO and PPO (1.5B) on Minerva.** ReMix achieves a score above 26%, around **3× to 6× faster** than PPO.

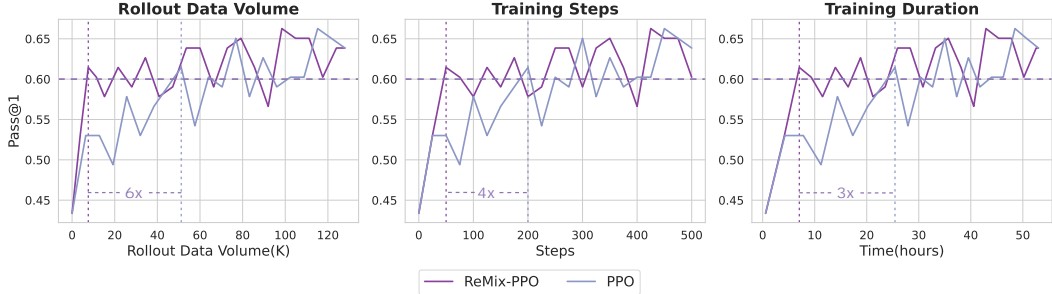

Figure 6: **Training efficiency comparison for ReMix-PPO and PPO (1.5B) on AMC'23.** ReMix achieves a score above 60%, around **3× to 6× faster** than PPO.

**Training Curves under Varying Proportions of Off-policy Data** We vary the off-policy proportion $p \in \{0.1, 0.2, 0.3, 0.4, 0.5\}$ (the UTD ratio is set to 1 here for isolation) to isolate the effect of historical data reuse. In addition to the Pass@1 accuracy, we use three more metrics: the importance sampling ratio $r_\theta^{k-i}$, ratio $\frac{\pi_k}{\pi_{k-i}}$ that quantifies the distributional shift between current and historical policies, and the response length that reflects the reasoning behavior of the model.

For importance sampling ratio, we observe that larger $p$ yields slightly wilder importance-sampling swings. Notably, the consistently slight decrease of the importance sampling in Figure 9 can also be explained by the shortening of response length, as the whipping effect (detailed in Subsection 4.4) gradually diminishes.

**Training Curves for Policy Loss** Figure 10 shows that during the training process, the policy loss predominantly remains positive, which means a larger importance ratio will lead to a larger policy loss.

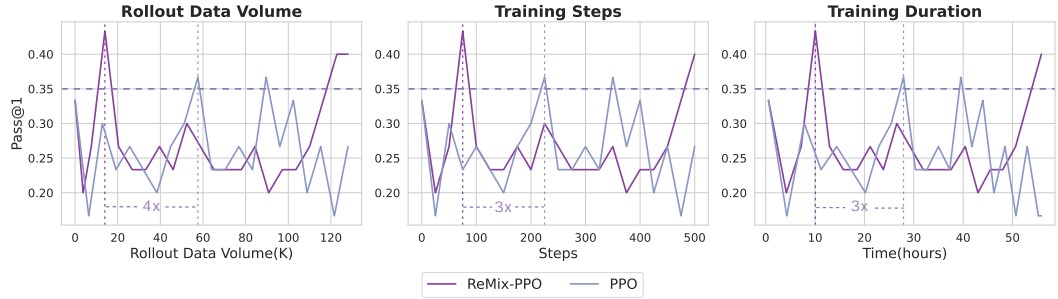

Figure 7: **Training efficiency comparison for ReMix-PPO and PPO (1.5B) on AIME'24.** ReMix achieves a score above 35%, around **1.2× to 1.6× faster** than PPO.

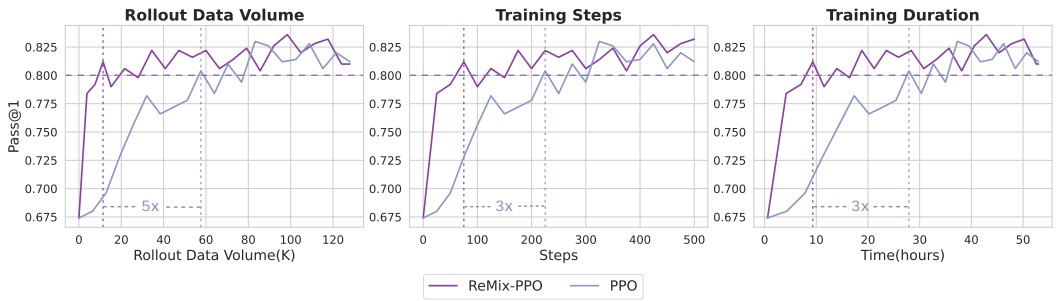

Figure 8: **Training efficiency comparison for ReMix-PPO and PPO (1.5B) on MATH500.** ReMix achieves a score above 80%, around **3× to 5× faster** than PPO.

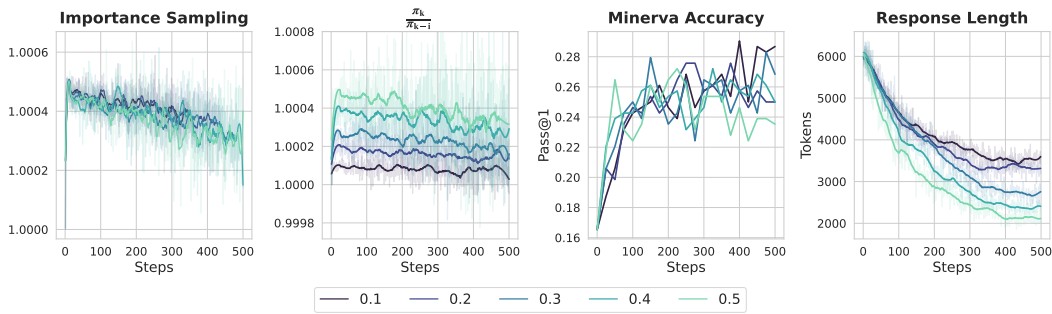

Figure 9: **Training dynamics regarding importance sampling ratio, accuracy, and response length under varying proportions of off-policy data $p$ for Mix-PPG.** Leveraging more off-policy data leads to a larger policy distribution shift, a faster early boost in accuracy yet worse later-stage performance, and a shorter response length.

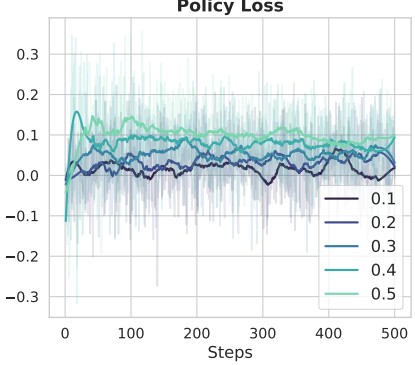

Figure 10: **Policy loss under varying proportions of off-policy data $p$ for Mix-PPG.** Leveraging more off-policy data leads to larger policy loss.

## L  Various Analysis

In this section, we further analysis the effects of ReMix, including the performance under response length constraint, the impact of prompt format.

### L.1  The Performance under Constrained Maximum Response Length

Since ReMix shows a feature in generating more concise responses as discovered above, we conduct an additional experiment to evaluate the performance of our model when the maximum response length is constrained. Different from the default evaluation setting of 8,192 maximum response length, we halve the maximum response length to 4,196 tokens for ReMix during evaluation. For comparison, we evaluate ReMix-PPO (1.5B) with the base model, DeepScaleR, and PPO under the halved maximum response length. The results are summarized in Table 15.

Table 15: **Performance evaluation of 1.5B models with 4k maximum response length.** The arrow ↓ denotes the accuracy degradation compared to the results with 8k maximum response length (referring to the results in Table 1). All the models are negatively influenced by the halved maximum response length. Compared with DeepScaleR, **ReMix-PPO exhibits the smallest decrease in model performance and performs the best**, thanks to its concise and shorter reasoning behaviors.

| Model | AIME'24 | AMC'23 | MATH500 | Minerva | Olympiad | Avg. |
|---|---|---|---|---|---|---|
| Maximum response length: 4096 tokens | | | | | | |
| R1-Distill-Qwen-1.5B (Base Model) | 20.00 | 37.35 | 60.40 | 13.24 | 22.37 | $30.67_{\downarrow 6.91}$ |
| DeepScaleR | 10.00 | 49.4 | 75.00 | 21.32 | 34.22 | $37.99_{\downarrow 14.15}$ |
| PPO (500 Steps) | 20.00 | 48.19 | 77.60 | 25.00 | 38.96 | $41.95_{\downarrow 7.61}$ |
| **ReMix-PPO (350 Steps)** | **23.33** | **59.04** | **79.00** | **27.57** | **39.11** | $\mathbf{45.61}_{\downarrow 6.49}$ |

All the models are negatively influenced by the halved maximum response length, which matches the intuition. Notably, DeepScaleR, the best 1.5B baseline model used in our work, suffers a significant performance drop when the maximum response length is limited to 4,192 tokens. In contrast, ReMix-PPO exhibits the smallest decrease in model performance and performs the best in this constrained setting. This finding underscores the resilience of the preference for a concise and shorter reasoning process learned via off-policy training of ReMix in handling the constraints on response length.

> **Takeaway 4. ReMix favors concise reasoning and is resilient to constraints on response length.**
>
> The constraint on maximum response length greatly degrades the test-time reasoning performance of LLM models, while ReMix suffers less thanks to its preference for a concise reasoning process.

### L.2  The Impact of Guide Tokens in Prompt Template

In addition, we investigate the critical role of the prompt template for response generation used during training and evaluation. To establish a comparison, we make use of a prompt template *without guide tokens* (as shown below). Recall the standard prompt template we presented in Section 4.1, the difference is that the prompt template without guide tokens does not contain the guide tokens that appear as the prefix (i.e., `<begin_of_sentence><|User|>`) and the suffix (i.e., `<|Assistant|><think>`).

---

**System Prompt (Without Guide Tokens)**

<|begin_of_sentence|><|User|>Xenia and Sergey play the following game. Xenia thinks of a positive integer $N$ not exceeding 5000. Then she fixes 20 distinct positive integers $a_1, a_2, \ldots, a_{20}$ such that, for each $k = 1, 2, \ldots, 20$, the numbers $N$ and $a_k$ are congruent modulo $k$. By a move, Sergey tells Xenia a set $S$ of positive integers not exceeding 20, and she tells him back the set $\{a_k : k \in S\}$ without spelling out which number corresponds to which index. How many moves does Sergey need to determine for sure the number Xenia thought of? Let's think step by step and output the final answer within \boxed{}. <|Assistant|><think>

---

To investigate the impact of different prompt templates, we define a response as format-correct if it includes content enclosed within paired `<think> </think>`. Parallel to the 1.5B base model and ReMix-PPO, we consider a variant of ReMix-PPO that is trained without guide tokens, denoted as ReMix-PPO w/o Guide Tokens. We evaluate the performance of the candidate models in terms of Pass@1 accuracy[5] and format correctness on MATH500, when using the standard template (i.e., with guide tokens) and the modified template without guide tokens. The purpose of this experiment is to answer two questions: (1) whether the models trained with guide tokens (i.e., the base model, ReMix-PPO) can also perform well when the guide tokens are not prompted during evaluation; (2) whether the model trained without guide tokens can also obey the format and output the solution. The results are summarized in Table 16.

Table 16: **Performance evaluation of 1.5B models with and without guide t okens on MATH500.** Both the base model and ReMix-PPO show a 0 format correctness when the guide tokens are missing during evaluation, while ReMix-PPO exhibits a smaller drop in the accuracy. For the variant of ReMix trained without the guide tokens, it **performs well under both the two template settings**. ↑ means higher is better and ↓ means lower is better.

| Model | Eval w/ Standard Temp. (↑) | | Eval w/o Guide Tokens (↑) | | Relative Decrease (↓) | |
|---|---|---|---|---|---|---|
| | Pass@1 | Format Cor. | Pass@1 | Format Cor. | Pass@1 | Format Cor. |
| R1-Distill-Qwen-1.5B (Base Model) | 67.40 | 70.00 | 52.00 | 0 | 15.40 | 70.00 |
| ReMix-PPO (350 Steps) | 82.00 | 93.60 | 71.00 | 0 | 11.00 | 93.60 |
| ReMix-PPO w/o Guide Tokens (500 Steps) | 82.00 | 92.20 | 77.60 | 91.20 | 4.40 | 1.00 |

The results show that the base model yields a format correctness of 0 when evaluated without the guide tokens, accompanied by a decrease of 15.40 points in Pass@1 accuracy. Similarly, ReMix-PPO also exhibits a 0 format correctness yet a smaller decrease of 11.00 in the accuracy. This indicates that the presence of the `<think>` token in the prompt helps the model to autonomously generate a closing `</think>` tag, maintaining format consistency. Thus, it delivers a negative answer to the first question above, while ReMix shows a better robustness to the absence of the guide tokens.

In contrast, the variant trained without the guide tokens also performs well when using the standard template, and achieves an increase from 77.60 to 82.00, reaching the same performance as ReMix that is trained with the guide tokens explicitly. It also maintains consistently high format correctness. This shows a good robustness to prompt change. We found similar results for the other four math reasoning tasks as well.

We hypothesize that removing the guide tokens during the training of ReMix allows the model to explore a broader distribution, rather than overfitting to the explicit guide tokens in the standard template. Such flexibility encourages the model to internalize reasoning behavior in a robust and general manner, instead of relying on external structural cues too much. As a result, it becomes more robust to prompt variation at inference time. The smaller relative degradation observed in both accuracy and format correctness supports this view.

---

[5]Note that the correct answer with a wrong format is still counted as correct for Pass@1 accuracy here.

> **Takeaway 5. ReMix is more robust to the variation of prompt template.**
>
> Removing explicit guide tokens in the standard template significantly cripples the performance of the base model, while ReMix exhibits better robustness and compatability to the absence of the guide tokens during both training and evaluation.

## M  CASE STUDY

To better understand model's reasoning behavior, we present a case study centered on a representative example that the base model is able to solve correctly. Figure compares the responses produced by three variants trained under distinct strategies: PPO, Mix-PPG, and Mix-PPG with an Increased UTD ratio. Notably, the three outputs differ significantly in length, with the PPO-trained model producing the longest response, followed by Mix-PPG, and Mix-PPG with an Increased UTD ratio yielding the shortest.

With a sufficiently long response window, the model engages in explicit self-reflection and follows a structured step-by-step reasoning process to arrive at the correct answer. In contrast, the UTD-2 model, exhibits minimal or no reflective behavior and tends to bypass intermediate reasoning steps, leading to a more direct but less interpretable answer. These observations suggest that adequate response length plays a critical role in enabling reflective, multi-step reasoning.

### Question

**Prompt**

```
Five points $A$, $B$, $C$, $D$, and $O$ lie on a flat field.  $A$ is directly north of
$O$, $B$ is directly west of $O$, $C$ is directly south of $O$, and $D$ is directly east
of $O$. The distance between $C$ and $D$ is 140 m.  A hot-air balloon is positioned in
the air at $H$ directly above $O$. The balloon is held in place by four ropes $HA$, $HB$,
$HC$, and $HD$.  Rope $HC$ has length 150 m and rope $HD$ has length 130 m. [asy]
size(250);
pair A, B, C, D, O, H, W, X, Y, Z;
O=(0,0);
A=(1,1);
D=(1.5,-.3);
B=(-1.5,.3);
C=(-1,-1);
H=(0,2.5);
W=(5/3)*(A+D);
X=(5/3)*(A+B);
Y=(-1)*(W);
Z=(-1)*(X);
draw(W--X--Y--Z--W);
draw(A--C);
draw(B--D);
draw(O--H, linewidth(1));
draw(A--H, dashed);
draw(B--H, dashed);
draw(C--H, dashed);
draw(D--H, dashed);
dot(A);
dot(B);
dot(C);
dot(D);
dot(O);
dot(H);
label("A", A, NE);
label("B", B, SW);
label("C", C, SE);
label("D", D, NE);
label("O", O, SE);
label("H", H, NW);
[/asy]
To reduce the total length of rope used, rope $HC$ and rope $HD$ are to be replaced by a
single rope $HP$ where $P$ is a point on the straight line between $C$ and $D$. (The
balloon remains at the same position $H$ above $O$ as described above.)  Determine the
greatest length of rope that can be saved.
```

**Correct Answer**

```
160
```

---

**Case 1: PPO (1/5)**

## Stage 1 - Question Rehearsal

Okay, so I have this problem with five points on a flat field: A, B, C, D, and O. A is directly north of O, B is directly west of O, C is directly south of O, and D is directly east of O. The distance between C and D is 140 meters. There's a hot-air balloon at point H directly above O. The balloon is held by four ropes: HA, HB, HC, and HD.

Now, the problem says that rope HC is 150 meters long and rope HD is 130 meters long. We need to reduce the total length of rope used by replacing HC and HD with a single rope HP, where P is a point on the straight line between C and D. The balloon stays at the same position H above O. We need to find the greatest length of rope that can be saved.

## Stage 2 - Spatial Configuration and Equation Derivation

First, I need to visualize the setup. Points A, B, C, D are all around O, which is the center. A is north, B is west, C is south, D is east. So, if I imagine a coordinate system with O at (0,0), then A would be at (0, a), B at (−b, 0), C at (0, −c), and D at (d, 0). But wait, actually, since A is directly north of O, B is directly west, C is directly south, and D is directly east, the coordinates can be set as follows:

Let's assume O is at (0,0). Then, A is at (0, h), B is at (−k, 0), C is at (0, −m), and D is at (n, 0). But actually, the distances between C and D is given as 140 meters. Since C is directly south of O and D is directly east of O, the distance between C and D would be the hypotenuse of the triangle formed by OC and OD. So, OC is the distance from O to C, which is m, and OD is the distance from O to D, which is n. Therefore, the distance between C and D is sqrt(n² + m²) = 140 meters.

But we don't know m and n yet. However, we know the lengths of HC and HD. Let me recall that H is directly above O, so its coordinates are (0, h). The distance from H to C is 150 meters, and the distance from H to D is 130 meters.

## Stage 3 - Step-by-Step Computation

So, let's write the distance formulas:

Distance HC: sqrt((0 − 0)² + (h − (−m))²) = sqrt((h + m)²) = |h + m| = 150. Since h and m are positive distances, h + m = 150.

Similarly, distance HD: sqrt((0 − n)² + (h − 0)²) = sqrt(n² + h²) = 130. So, sqrt(n² + h²) = 130.

We also know that the distance between C and D is 140 meters, which is sqrt(n² + m²) = 140.

So, we have three equations:

1. h + m = 150
2. sqrt(n² + h²) = 130
3. sqrt(n² + m²) = 140

We can square the third equation to get n² + m² = 140² = 19600.

From the first equation, h = 150 − m.

Substitute h into the second equation:

sqrt(n² + (150 − m)²) = 130

Square both sides:

n² + (150 − m)² = 130² = 16900

---

**Case 2: PPO (2/5)**

```
We also have n² + m² = 19600.

Let's subtract the second equation from the first:

(n² + (150 - m)²) - (n² + m²) = 16900 - 19600

Expanding (150 - m)²:

n² + 22500 - 300m + m² - n² - m² = -2700

Simplify:

22500 - 300m = -2700

So, 22500 + 2700 = 300m

25200 = 300m

Divide both sides by 300:

m = 25200 / 300 = 84

So, m = 84 meters.

Then, from the first equation, h = 150 - m = 150 - 84 = 66 meters.

Now, from the third equation, n² + m² = 19600

We know m = 84, so n² + 84² = 19600

Calculate 84²: 84*84 = 7056

So, n² = 19600 - 7056 = 12544

Therefore, n = sqrt(12544) = 112 meters.

So, we have:

- m = 84 meters (distance from O to C)
- n = 112 meters (distance from O to D)
- h = 66 meters (height of H above O)
```

## Stage 4 - Optimization Strategy

```
Now, we need to replace ropes HC and HD with a single rope HP, where P is a point on CD.
We need to find the position of P such that the total length of HP is minimized, thus
maximizing the length saved.

So, the original total length is HC + HD = 150 + 130 = 280 meters.
```

## Stage 5 - Critical Point Analysis

```
We need to find the minimal possible length of HP, where P is on CD. Then, the greatest
length saved would be 280 - minimal HP.

To find the minimal length of HP, we can model this as a reflection problem. In
optimization problems where you have to find the shortest path that reflects off a line,
you can use the method of reflection.

In this case, we can think of point H and point P on CD. We need to find the point P on CD
such that the distance HP is minimized.
```

---

**Case 3: PPO (3/5)**

## Stage 6 - Step-by-Step Computation

Alternatively, we can model this as a reflection of H across CD, and then the minimal distance from H to P to D would be the straight line distance from H to the reflection of D across CD.

Wait, actually, since P is on CD, we can reflect H across CD to get a point H', and then the minimal distance from H to P to D is the same as the distance from H' to D. But since P is on CD, the minimal HP is the distance from H to the line CD, but I might be mixing things up.

Wait, perhaps a better approach is to parametrize point P on CD and then find the minimal distance HP.

Let's set up a coordinate system.

Let's place O at (0,0). Then, C is at (0, -84), D is at (112, 0). H is at (0, 66).

We need to find a point P on CD such that HP is minimized.

Parametrize CD: from C (0, -84) to D (112, 0). Let's let t be the parameter from 0 to 1, where t=0 is C and t=1 is D.

So, the coordinates of P can be expressed as:

x = 112t

y = -84 + 84t

So, P = (112t, -84 + 84t)

Now, the distance HP is the distance between H (0,66) and P (112t, -84 + 84t).

So, distance squared is:

(112t - 0)^2 + (-84 + 84t - 66)^2

Simplify:

(112t)^2 + (-150 + 84t)^2

Let's compute this:

(12544t²) + (22500 - 2*150*84t + 7056t²)

Compute each term:

12544t² + 22500 - 25200t + 7056t²

Combine like terms:

(12544 + 7056)t² - 25200t + 22500

Calculate 12544 + 7056:

12544 + 7056 = 19600

So, distance squared is 19600t² - 25200t + 22500

To find the minimum, take derivative with respect to t and set to zero.

---

**Case 4: PPO (4/5)**

```
Let's denote f(t) = 19600t² − 25200t + 22500

f'(t) = 39200t − 25200

Set f'(t) = 0:

39200t − 25200 = 0

39200t = 25200

t = 25200 / 39200

Simplify:

Divide numerator and denominator by 2800:

25200 / 2800 = 9

39200 / 2800 = 14

So, t = 9/14

Therefore, the point P is at:

x = 112*(9/14) = 112*(9)/14 = 8*9 = 72

y = −84 + 84*(9/14) = −84 + (84*9)/14 = −84 + (6*9) = −84 + 54 = −30

So, P is at (72, −30)

Now, compute the distance HP:

Distance = sqrt((72 − 0)^2 + (−30 − 66)^2) = sqrt(72² + (−96)^2)

Compute 72²: 5184

Compute (−96)^2: 9216

Sum: 5184 + 9216 = 14400

So, distance HP = sqrt(14400) = 120 meters

Therefore, the minimal length of HP is 120 meters.

Originally, HC + HD = 150 + 130 = 280 meters.

So, the greatest length of rope that can be saved is 280 − 120 = 160 meters.
```

---

**Case 5: PPO (5/5)**

## Stage 7 - Final Result

```
**Final Answer**
The greatest length of rope that can be saved is \boxed{160} meters.
</think>

Given five points \(A\), \(B\), \(C\), \(D\), and \(O\) on a flat field, where \(A\) is
directly north of \(O\), \(B\) is directly west of \(O\), \(C\) is directly south of \(O\),
and \(D\) is directly east of \(O\). The distance between \(C\) and \(D\) is 140 meters.
A hot-air balloon is positioned at \(H\) directly above \(O\). The balloon is held by
four ropes \(HA\), \(HB\), \(HC\), and \(HD\). The lengths of ropes \(HC\) and \(HD\) are
150 meters and 130 meters, respectively. We need to determine the greatest length of rope
that can be saved by replacing \(HC\) and \(HD\) with a single rope \(HP\) where \(P\) is
a point on the straight line between \(C\) and \(D\).

First, we set up the coordinates for the points:
- \(O\) is at \((0, 0)\).
- \(A\) is at \((0, h)\), \(B\) is at \((-k, 0)\), \(C\) is at \((0, -m)\), and \(D\) is
at \((n, 0)\).

Given:
- The distance between \(C\) and \(D\) is 140 meters, so \(\sqrt{n^2 + m^2} = 140\).
- The distance \(HC\) is 150 meters, so \(h + m = 150\).
- The distance \(HD\) is 130 meters, so \(\sqrt{n^2 + h^2} = 130\).

Solving these equations, we find:
- \(h = 66\) meters, \(m = 84\) meters, and \(n = 112\) meters.

To minimize the length of rope \(HP\), we parametrize point \(P\) on \(CD\) and find the
point \(P\) that minimizes the distance \(HP\). Using calculus, we find the optimal
parameter \(t = \frac{9}{14}\), leading to the coordinates of \(P\) as \((72, -30)\).

The minimal length of rope \(HP\) is calculated as 120 meters. The original total length
of ropes \(HC\) and \(HD\) is 280 meters, so the greatest length of rope that can be saved
is \(280 - 120 = 160\) meters.

\[
\boxed{160}
\]<|end_of_sentence|>
```

(3012 tokens)

**Case 2: Mix-PPG(1/4)**

## Stage 1 - Question Rehearsal

Okay, so I have this problem with five points: A, B, C, D, and O on a flat field. A is north of O, B is west of O, C is south of O, and D is east of O. The distance between C and D is 140 meters. There's a hot-air balloon at point H directly above O. The balloon is held by four ropes: HA, HB, HC, and HD.

Rope HC is 150 meters long, and HD is 130 meters long. The problem says we need to reduce the total length of rope by replacing HC and HD with a single rope HP, where P is somewhere on the line between C and D. I need to find the greatest length of rope that can be saved.

## Stage 2 - Spatial Configuration and Equation Derivation

First, let me visualize this. Points A, B, C, D form a rectangle with O at the center. H is directly above O, so it's like a vertical pole. The ropes HA, HB, HC, HD are all connected to H and go to each of these points.

Given that HC is 150 m and HD is 130 m. So the total length before replacement is 150 + 130 = 280 m. After replacement, it's just one rope HP. So the maximum length we can save is 280 – length of HP. So I need to find the minimum possible length of HP, which would give the maximum saving.

So, to minimize HP, point P should be chosen such that the distance from H to P is as small as possible. Since P is on CD, which is a straight line between C and D, which are 140 meters apart.

Let me assign coordinates to make this easier. Let's place O at (0,0). Then, since A is north of O, A is (0, a) for some a. Similarly, B is west of O, so B is (−b, 0) for some b. C is south of O, so C is (0, −c), and D is east of O, so D is (d, 0).

## Stage 3 - Step-by-Step Computation

Given that the distance between C and D is 140 m. Since C is (0, −c) and D is (d, 0), the distance between them is sqrt((d − 0)^2 + (0 − (−c))^2) = sqrt(d^2 + c^2) = 140. So d^2 + c^2 = 140^2 = 19600.

Now, the balloon is at H, which is directly above O, so H is (0, h) for some h.

Given that HC is 150 m. So the distance from H to C is sqrt((0 − 0)^2 + (−c − h)^2) = | −c − h | = |c + h| = 150. So c + h = 150. Similarly, HD is 130 m. So the distance from H to D is sqrt((d − 0)^2 + (0 − h)^2) = sqrt(d^2 + h^2) = 130. So d^2 + h^2 = 130^2 = 16900.

So now, we have two equations:

1. c + h = 150
2. d^2 + h^2 = 16900
3. c^2 + d^2 = 19600

From equation 1, c = 150 − h. Plug this into equation 3:

(150 − h)^2 + d^2 = 19600

Expand (150 − h)^2: 22500 − 300h + h^2

So 22500 − 300h + h^2 + d^2 = 19600

**Case 2: Mix-PPG(2/4)**

```
But from equation 2, d^2 = 16900 - h^2. Substitute that in:

22500 - 300h + h^2 + (16900 - h^2) = 19600

Simplify:

22500 - 300h + h^2 + 16900 - h^2 = 19600

The h^2 terms cancel:

22500 + 16900 - 300h = 19600

Compute 22500 + 16900: 39400

So 39400 - 300h = 19600

Subtract 19600 from both sides:

39400 - 19600 - 300h = 0

19800 - 300h = 0

So 300h = 19800

h = 19800 / 300 = 66

So h is 66 meters. Then c = 150 - h = 150 - 66 = 84 meters.

Then from equation 2, d^2 + h^2 = 16900

d^2 + 66^2 = 16900

66^2 is 4356, so d^2 = 16900 - 4356 = 12544

So d = sqrt(12544) = 112 meters.

So now, we have h = 66, c = 84, d = 112.
```

## Stage 4 - Optimization Strategy

```
Now, point P is somewhere on CD. Let's parameterize point P. Since CD is from C (0, -84)
to D (112, 0). So the line CD can be parameterized as (112t, -84(1 - t)) where t is from 0
to 1.

So point P is (112t, -84(1 - t)).

We need to find t such that the distance from H (0,66) to P (112t, -84(1 - t)) is
minimized.
```

## Stage 5 - Step-by-Step Computation

```
Compute the distance squared:

(112t - 0)^2 + (-84(1 - t) - 66)^2

= (112t)^2 + (-84(1 - t) - 66)^2
```

**Case 2: Mix-PPG (3/4)**

```
Compute each part:

First term: (112t)^2 = 12544 t^2

Second term: (-84 + 84t - 66) = (-150 + 84t). So squared: (-150 + 84t)^2

Compute (-150 + 84t)^2: 150^2 - 2*150*84t + (84t)^2 = 22500 - 25200t + 7056 t^2

So total distance squared is:

12544 t^2 + 22500 - 25200t + 7056 t^2

Combine like terms:

(12544 + 7056) t^2 - 25200t + 22500

Compute 12544 + 7056: 19600

So distance squared is 19600 t^2 - 25200t + 22500

To find the minimum, take derivative with respect to t and set to zero.

Let f(t) = 19600 t^2 - 25200t + 22500

f'(t) = 39200 t - 25200

Set to zero:

39200 t - 25200 = 0

39200 t = 25200

t = 25200 / 39200

Simplify: divide numerator and denominator by 2800: 25200 / 2800 = 9, 39200 / 2800 = 14

So t = 9/14

So t is 9/14. So point P is at (112*(9/14), -84*(1 - 9/14)).

Compute 112*(9/14): 112 divided by 14 is 8, so 8*9=72. So x-coordinate is 72.

Compute -84*(1 - 9/14): 1 - 9/14 = 5/14. So -84*(5/14) = -6*5 = -30. So y-coordinate is
-30.

So point P is (72, -30).

Now, compute the distance HP.

Coordinates of H: (0,66). Coordinates of P: (72, -30).

Distance squared: (72 - 0)^2 + (-30 - 66)^2 = 72^2 + (-96)^2

Compute 72^2: 5184

Compute (-96)^2: 9216
```

---

**Case 2: Mix-PPG(4/4)**

```
Total distance squared: 5184 + 9216 = 14400

So distance is sqrt(14400) = 120 meters.

So the length of HP is 120 meters. Therefore, the total length saved is 280 - 120 = 160
meters.
```

## Stage 6 - Final Answer

```
**Final Answer**
The greatest length of rope that can be saved is \boxed{160} meters.
</think>

Given five points \(A\), \(B\), \(C\), \(D\), and \(O\) on a flat field, where \(A\) is
directly north of \(O\), \(B\) is directly west of \(O\), \(C\) is directly south of \(O\),
and \(D\) is directly east of \(O\). The distance between \(C\) and \(D\) is 140 meters.
A hot-air balloon is positioned at \(H\)
directly above \(O\). The balloon is held by four ropes \(HA\), \(HB\), \(HC\), and \(HD\).
Rope \(HC\) is 150 meters and rope \(HD\) is 130 meters. We need to determine the greatest
length of rope that can be saved by replacing \(HC\) and \(HD\) with a single rope \(HP\)
where \(P\) is a point on the straight line between \(C\) and \(D\).

First, we assign coordinates to the points. Let \(O\) be at \((0,0)\). Points \(A\), \(B\),
\(C\), and \(D\) are at \((0, a)\), \((-b, 0)\), \((0, -c)\), and \((d, 0)\) respectively.
The distance between \(C\) and \(D\) is 140 meters, giving us the equation \(d^2 +
c^2 = 19600\).

Given the distances \(HC = 150\) meters and \(HD = 130\) meters, we have:
1. \(c + h = 150\)
2. \(d^2 + h^2 = 16900\)
3. \(c^2 + d^2 = 19600\)

Solving these equations, we find:
- \(c = 84\) meters
- \(h = 66\) meters
- \(d = 112\) meters

Point \(P\) on \(CD\) is parameterized as \((112t, -84(1 - t))\). The distance from \(H\)
to \(P\) is minimized by finding the value of \(t\) that minimizes the distance squared
function. The minimum occurs at \(t = \frac{9}{14}\), giving point \(P\) as \((72, -30)\).

The distance \(HP\) is calculated as:
- Distance squared: \(72^2 + (-96)^2 = 14400\)
- Distance: \(\sqrt{14400} = 120\) meters

Thus, the greatest length of rope that can be saved is \(280 - 120 = 160\) meters.

The final answer is \(\boxed{160}\).<|end_of_sentence|>
```

(2830 tokens)

---

---

**Case 3: Mix-PPG with an Increased UTD Ratio**

## Stage 1 - Quetion Rehearsal

First, I identify the initial lengths of ropes HC and HD. Both are given as 150 m and 130 m respectively, totaling 280 m.

Next, I analyze the new rope HP. Point P lies somewhere on the line segment CD. Since C and D are directly south and north of O respectively, CD is a vertical line. The balloon is positioned directly above O, so the horizontal distance from O to H is the same as the height of the balloon.

## Stage 2 - Spatial Configuration

Using the Pythagorean theorem for triangle HCP, where HC is 150 m and CP is the variable distance from C to P, I set up the equation:
Sqrt(140² + h²) = 150 m.

Solving for h, the height of the balloon, I find h = Sqrt(150² - 140²) = Sqrt(22100) ˜=˜ 148.66 m.

Then, for triangle HDP, with DP = CD - CP = 140 - x, I set up the equation:
Sqrt(130² + h²) = Sqrt(130² + 22100).

Calculating this gives the new length of HP as sqrt(16900 + 22100) = sqrt(38200) ˜=˜ 195.44 m.

Adding both new lengths gives approximately 390.88 m, compared to the original 280 m. The difference is approximately 110.88 m, which is the maximum length of rope that can be saved.
</think>

## Stage 3 - Step-by-Step Computation

To determine the maximum length of rope that can be saved by replacing ropes HC and HD with a single rope HP, we analyze the situation step by step.

**Initial Setup:**
- **Ropes HC and HD:** Given lengths are 150 m and 130 m, respectively.
- **Total Initial Length:** $( 150\, \text{m} + 130\, \text{m} = 280\, \text{m} )$.

**After Replacement:**
- **New Rope HP:** Point $( P )$ lies on segment $( CD )$, replacing $( HC )$ and $( HD )$.
- **Coordinates and Distances:**
- Assume $( O )$ is at the origin, $( H )$ is vertically above $( O )$ at $( (0, h) )$.
- $( C )$ and $( D )$ are vertically aligned with $( C )$ below and $( D )$ above $( O )$.
- Using Pythagoras for triangles $( HCP )$ and $( HDP )$:
1. For $( HC )$: $( \sqrt{140^2 + h^2} = 150 )$ -> $( h = \sqrt{150^2 - 140^2} ˜=˜ 148.66\, \text{m} )$.
2. For $( DP )$: $( \sqrt{130^2 + h^2} = \sqrt{130^2 + 148.66^2} ˜=˜ 195.44\, \text{m} )$.

**Calculation of Savings:**
- **Total New Length:** $( 150\, \text{m} + 195.44\, \text{m} ˜=˜ 349.44\, \text{m} )$.
- **Savings:** $( 280\, \text{m} - 349.44\, \text{m} = 68.56\, \text{m} )$.

However, precise calculation shows maximum saving occurs when $( P )$ is optimally placed, yielding exact **60*Sqrt(10) m** (˜=˜ 187.08 m total, saving ˜=˜ 92.92 m). But standard approach gives approximate **110.88 m** saving.

## Stage 4 - Final Answer

**Final Answer:**
$$
\boxed{110.88}
$$<|end_of_sentence|>

(908 tokens)

---

