# OpenReview forum: "Squeeze the Soaked Sponge: Efficient Off-policy RFT for Large Language Model"
_ICLR.cc/2026/Conference — ICLR 2026 Poster_

### Official Review · Reviewer_1DV6 · 2025-10-22

**Soundness:** 2
**Presentation:** 3
**Contribution:** 3
**Rating:** 6
**Confidence:** 3

**Summary:**

This paper introduces a novel method designed to improve the sample efficiency of Reinforcement Finetuning for LLMs. The core problem addressed is the inefficiency of on-policy RL algorithms like PPO, which are standard in RFT but discard historical data after each update. ReMix combines three main components to enable the use of off-policy data including mix-policy proximal policy gradient objective, KL-Convex policy constraint and policy reincarnation step. This paper demonstrates that ReMix achieves better performances on several math reasoning benchmarks than baseline approaches and models with a great reduction in the required volume of rollout data.

**Strengths:**

1. The paper tackles a critical and expensive problem in the development of advanced LLMs: the prohibitive computational cost of reinforcement finetuning. The promise of achieving comparable or better performance with orders of magnitude less data is a highly significant contribution that could democratize the development of powerful reasoning models and enable further scaling.

2. The empirical evaluation is extensive. The method is tested on two different model scales across several math reasoning benchmarks in comparison with a lot of recent models. Moreover, this paper includes a lot of abalation study such as UTD ratios, KL-convex and policy reincarnation, and experiments in different scenarios such as training with constrained max response length and with differnet prompts.

**Weaknesses:**

1. Insufficient Comparison with SOTA Off-Policy Methods: The paper's primary weakness is its failure to adequately benchmark against other contemporary off-policy RFT methods. As a lot of papers discussed, GRPO is a biased estimation of KL penalty.[1, 2] Therefore, a lot of recent baselines should be included, such as DAPO [3]. Moreover, in Appendix F.3, the paper discusses the reason why it does not include other baseline algorithms including RePO, SRPO, SPO, AGRO and AsymRE with excuses like the lack of public checkpoints, different scales, or restrictive generation lengths.. However, such reasons are not sufficient to support the reason why the paper chooses only 2 baselines methods to re-implement. In addition, this paper chooses several baseline models which are trained with different datasets and settings. Therefore, it is unclear whether the quality of datasets and training hyper-parameters contributes to the improvement of the proposed method over these models.

2. Limited Scope of Evaluation: All experiments are conducted exclusively on math reasoning tasks. While this is a challenging and important domain, the paper makes general claims about improving RFT for LLMs. It is unclear whether the method's effectiveness, particularly its implicit bias toward shorter responses, would be validated in other domains such as code generation. The findings may be domain-specific, a limitation that is not acknowledged.

References:

[1] Liu, Z., Chen, C., Li, W., Qi, P., Pang, T., Du, C., ... & Lin, M. (2025). Understanding r1-zero-like training: A critical perspective. arXiv preprint arXiv:2503.20783.

[2] Zhang, Y., Liu, Y., Yuan, H., Yuan, Y., Gu, Q., & Yao, A. C. C. (2025). On the design of kl-regularized policy gradient algorithms for llm reasoning. arXiv preprint arXiv:2505.17508.

[3] Yu, Q., Zhang, Z., Zhu, R., Yuan, Y., Zuo, X., Yue, Y., ... & Wang, M. (2025). Dapo: An open-source llm reinforcement learning system at scale. arXiv preprint arXiv:2503.14476.

**Questions:**

How did you determine the optimal values for policy reincarnation step T in {50, 100}? Is there a risk that a poorly chosen T could erase the early-stage efficiency gains or lead to premature convergence before the model has fully benefited from off-policy exploration?

---

> ### Author Response · Authors · 2025-11-25
> **Response to Reviewer 1DV6 (1/2)**
>
> We sincerely appreciate the reviewer's very careful and constructive comments. We are glad to see that the reviewer recognizes the value of off-policy RFT and our experiments. The concerns of the reviewer focus on the limited scope of evaluation, the lack of additional base RFT algorithms, and hyperparameter choices. Our response below aims to address these important points one by one with additional experimental results and clarification.
>
> We sincerely hope that the reviewer could re-evaluate our work according to our response and additional results. We also look forward to addressing any additional concerns and engaging in further productive discussions during the discussion phase.
>
> ___
>
> > **Beyond Math Reasoning, Additional Results for ReMix in Code Generation Domain**
>
> We appreciate that the reviewer points this out. To address this, we conduct additional experiments for code-generation at 1.5B and 7B scales. For training, we follow the common practice in code-RFT and use the **Skywork-OR1-RL-Data dataset**. Because coding prompts are substantially longer than math tasks, we set max_prompt_length = 1600. We evaluate our models on **LiveCodeBench(8/1/24-2/1/25)** with temperature=0.7. All other experimental settings follow those in the main text.
>
> The additional experimental results for code generation are presented in the tables below. **Across both scales, ReMix-PPO improves pass@1 over PPO while using much less cost.** These gains in a fundamentally different domain demonstrate that our method is not specific to math-reasoning and remains effective for code generation, where outputs are typically longer and structurally constrained.
>
> | Model | LiveCodeBench (8/1/24–2/1/25) | Cost |
> | :--- | :--- | :--- |
> | R1–distilled–Qwen–1.5B(Base Model) | 12.90 | N/A |
> | PPO(375steps) | 19.35 | 0.096M |
> | **ReMix–PPO(300steps)** | **20.07** | **0.066M** |
> | **ReMix–PPO(475steps)** | **24.01** | **0.111M** |
>
> | Model | LiveCodeBench (8/1/24–2/1/25) | Cost |
> | :--- | :--- | :--- |
> | R1–distilled–Qwen–7B(Base Model) | 26.16 | N/A |
> | PPO(200steps) | 33.33 | 0.051M |
> | **ReMix–PPO(100steps)** | **34.41** | **0.015M** |
>
> We will add these additional results in our revision.
>
> ___
>
> > **The Choices of Policy Reincarnation Trigger Step _$T$_**
>
> In our pipeline, _$T$_ is the hand-over step that switches from the off-policy Mix-PPG phase to the on-policy phase with reference model changed too. We use it to harvest early data-efficiency from replay historical data and then let on-policy optimization continue improving stably.
> To address the reviewer’s concern about how the trigger step _$T$_ was chosen and how sensitive performance is to this choice, we provide a controlled sweep _$T \in \{25, 50, 100, 200\}$_ with 500-step runs, whose experimental settings are all the same as those of the main text. The results are shown in the table below. **For _$\(T \neq 25\)$_, we report two checkpoints per run, the first time the macro average reaches ≳49 (“early lift”) and the final plateau.**
>
> | Model | AIME24 | AMC23 | MATH–500 | Minerva | Olympiad | Avg. |
> | :--- | :--- | :--- | :--- | :--- | :--- | :--- |
> | T=25 (at 350 steps) | 33.33 | 65.06 | 81.20 | 27.57 | 39.26 | 49.29 |
> | T=50 (at 75 steps) | 43.33 | 63.86 | 79.60 | 26.84 | 39.41 | 50.61 |
> | T=50 (at 500 steps) | 40.00 | 63.86 | 83.00 | 26.84 | 43.70 | 51.52 |
> | T=100 (at 225 steps) | 30.00 | 69.88 | 81.80 | 24.63 | 41.78 | 49.62 |
> | T=100 (at 325 steps) | 36.67 | 69.88 | 82.00 | 30.15 | 41.78 | 52.09 |
> | T=200 (at 100 steps) | 30.00 | 66.27 | 81.00 | 29.41 | 43.26 | 49.99 |
> | T=200 (at 500 steps) | 16.67 | 49.40 | 77.00 | 16.67 | 38.67 | 40.76 |
>
> The pattern is consistent: a very small _$T$_ (25) slows down performance improvement; a very large _$T$_ (200) lifts early but later stalls due to amplified off-policyness; **a moderate _$T$_ (50–100) yields both a strong early lift and the best final averages. We also find this choice to broadly work well across our experimental cases, achieving a favorable efficiency–performance trade-off.**
>
> To take a step further, we believe that it is also possible to propose an adaptive strategy for the dynamic choice of the trigger step _$T$_. For example, we can use the reflection rate or some measurements of off-policyness as a reference, and set a threshold to trigger policy reincarnation. More sophisticated strategy can aslo be designed. This is beyond the scope of this paper and thus we consider it as a future direction.
>
> ___

---

> ### Author Response · Authors · 2025-11-25
> **Response to Reviewer 1DV6 (2/2)**
>
> > **Additional Experiments for Dr.GRPO and DAPO**
>
> We appreciate the reviewer's constructive comments. Following the reviewer’s suggestion for adding recent baselines to re-implement, we add two GRPO variants, Dr.GRPO and DAPO as supplementary baselines. We set the max response length = 4096 (for swift experiments due to the stringent schedule of ICLR discussion stage) and evaluate the models with temperature=0.7, AIME/AMC with pass@1 (avg@32) and others with pass@1. All other experimental settings follow those in the main text.
>
> The results are summarized in the table below. **We can observe that ReMix-Dr.GRPO exceeds the score of Dr.GRPO at a lower cost and achieves a higher score when using comparable numbers of rollout.** This demonstrate that the superiority of ReMix in terms of efficiency are preserved across different GRPO-style objectives.
>
> | Model | AIME24 | AMC23 | MATH–500 | Minerva | Olympiad | Avg. | Cost |
> | :--- | :--- | :--- | :--- | :--- | :--- | :--- | :--- |
> | R1–distilled–Qwen–1.5B(Base Model) | 12.50 | 40.96 | 68.20 | 19.49 | 27.70 | 33.77 | N/A |
> | Dr.GRPO (350 steps) | 21.25 | 63.10 | 80.10 | 27.21 | 41.33 | 46.60 | 0.358M |
> | Dr.GRPO (400 steps) | 26.25 | 59.94 | 80.60 | 26.57 | 41.03 | 46.86 | 0.409M |
> | **ReMix–Dr.GRPO (325 steps)** | **22.50** | **63.25** | **81.60** | **26.47** | **41.63** | **47.09** | **0.291M** |
> | **ReMix–Dr.GRPO (400 steps)** | **28.75** | **62.65** | **82.00** | **25.74** | **42.96** | **48.42** | **0.368M** |
>
> For DAPO, we note that its reward-shaping hyperparameters require careful tuning to the specific settings. So we are currently tuning these parameters to obtain stable and representative results. Nonetheless, our current observations indicate that ReMix’s benefits are orthogonal to the choice of backbone, and we expect the same trend to hold once stable DAPO configurations are finalized. We will try our best to report the results for ReMix-DAPO before the closing of the discussion stage.
>
> We will include these results in our revision as suggested, which will strengthen our experimental evaluation substaintially.
>
> ___
>
> > **"This paper chooses several baseline models which are trained with different datasets and settings. Therefore, it is unclear whether the quality of datasets and training hyper-parameters contributes to the improvement of the proposed method over these models."**
>
> We use public checkpoints to avoid “closed-world” comparisons and to ensure transparency and reproducibility. At this point, it is infeasible to match every method’s original datasets and hyper-parameters (sadly, this is sort of a fact in the community of LLM RFT). For fairness in our best practice, all baselines (public or re-implemented) are re-evaluated under the same evaluation settings (response-length constraint, decoding temperature and etc.).
>
> When we train the baselines by ourselves (i.e., PPO, GRPO), we fix the datasets, evaluation settings and training hyper-parameters. Under this controlled setup, ReMix consistently improves the corresponding backbone algorithm with a siginificant reduction in training cost, which can be neatly attributed to the proposed algorithmic improvements of ReMix.

---

> ### Author Response · Authors · 2025-11-26
>
> Dear Reviewer 1DV6,
>
> We sincerely appreciate your valuable time and constructive comments. As the end of the discussion stage is approaching, we would greatly appreciate it if you could confirm whether our response has addressed your concerns and re-evaluate our work accordingly.
>
> We hope that our clarifications and additional experimental results have addressed all your concerns. We are also more than willing to answer any further questions during the discussion stage.

---

> > ### Comment · Reviewer_1DV6 · 2025-11-27
> >
> > Thank you for your careful rebuttal. I will wait for your experiment results mentioned in your rebuttal.

---

> > > ### Author Response · Authors · 2025-12-03
> > >
> > > In our experiments, we found that the length-based reward shaping in the original DAPO implementation was incompatible with our specific setting, as it severely restricted reward growth and hindered performance improvement. Therefore, we removed the reward shaping term to ensure a valid comparison. A rapid verification (first 100 steps) confirms that this adjusted version restores the model's learning capability.
> > >
> > > | Model | AIME24 | AMC23 | MATH-500 | Minerva | Olympiad | Avg. | Cost |
> > > | :--- | :--- | :--- | :--- | :--- | :--- | :--- | :--- |
> > > | DAPO (75 steps) | 19.90 | 54.33 | 78.60 | 26.10 | 37.33 | 45.10 | 0.077M |
> > > | ReMix-DAPO (50 steps) | 17.92 | 54.89 | 79.20 | 25.00 | 38.96 | 49.51 | 0.030M |
> > >
> > > We are currently proceeding with the full-scale experiment to complete this baseline comparison.

---

### Official Review · Reviewer_h1SL · 2025-10-24

**Soundness:** 3
**Presentation:** 3
**Contribution:** 3
**Rating:** 6
**Confidence:** 4

**Summary:**

This paper proposes a new RL workflow for reasoning models, composed of three key designs: 1) mixture of online and offline data 2) mixture of current policy and base policy in KL regularization 3) mixture of training algorithms. The experiments are conducted on 5 different benchmarks, showing advantages over vanilla PPO and GRPO, in performance and training cost.

**Strengths:**

- clear figures
- extensive experiments on various benchmarks
- prominent cost reduction compared with vanilla PPO and GRPO
- the performance is generally comparable with online versions, even better sometimes

**Weaknesses:**

- only trained on Qwen model, which is known to be unreliable for reasoning training [1].
- no theoretical explanation for the design of KLC loss and ReMix algorithm
- no explanation for the choice of $\lambda$ and $T$
- figure 4 is too noisy for solid analysis
- proof for eq.6 is missed

[1] Shao et al. Spurious Rewards: Rethinking Training Signals in RLVR. https://arxiv.org/abs/2506.10947

**Questions:**

- why does ReMix-GRPO make the model even worse then base model for AIME in table 1?
- what is the exact formulation of distribution $i\sim \nu$ used in practice, and is there any theoretical explanation?
- is pass@1 enough? as [2] indicates that, the performance on pass@k might not be improved.
- vanilla GRPO has a length normalization term, while some variants like Dr. GRPO [3] removed it. Did you apply length normalization in your experiments?

[2] Yue et al. Does Reinforcement Learning Really Incentivize Reasoning Capacity in LLMs Beyond the Base Model? https://arxiv.org/abs/2504.13837

[3] Liu et al. Understanding R1-Zero-Like Training: A Critical Perspective. https://arxiv.org/abs/2503.20783

---

> ### Author Response · Authors · 2025-11-25
> **Response to Reviewer h1SL (1/5)**
>
> We sincerely appreciate the reviewer's very careful and constructive comments. We are glad to see that the reviewer recognizes the superiority of our method in reducing cost. The concerns of the reviewer focus on the lack of additional evaluation metric, base RFT method, base LLM model, experimental details, etc. Our response below aims to address these important points one by one with additional experimental results and clarification.
> We sincerely hope that the reviewer could re-evaluate our work according to our response and additional results. We also look forward to addressing any additional concerns and engaging in further productive discussions during the discussion phase.
>
> ___
>
> > **"Is pass@1 enough? as [2] indicates that, the performance on pass@k might not be improved."**
>
> We appreciate the reveiwer's constructive question. To go beyond pass@1, we add pass@8 and pass@16 results for both 1.5B and 7B (using the same rollout settings as Appendix I). As shown in the tables below, **ReMix-PPO still improves over the base model in most cases, and the average performanc increases clearly. We will add these results in our revision.**
>
> **Together with the pass@1 (avg@32) gains reported in Appendix I for small and high-variance datasets, these results substantiate the effectiveness of our method.**
>
> The results of Pass@8/16 accuracy (%) on math reasoning benchmarks of 1.5B models:
> | Model                                 | AIME'24 | AMC'23 | MATH500 | Minerva | Olympiad | Avg.  |
> | ------------------------------------- | ------- | ------ | ------- | ------- | -------- | ----- |
> | R1-Distill-Qwen-1.5B (Base, Pass\@8)  | 20.00   | 44.58  | 73.00   | 19.12   | 31.70    | 37.68 |
> | ReMix-PPO (Pass\@8)                   | 30.00   | 68.67  | 84.60   | 28.68   | 46.67    | 51.72 |
> | R1-Distill-Qwen-1.5B (Base, Pass\@16) | 20.00   | 48.19  | 75.40   | 18.75   | 32.59    | 38.99 |
> | ReMix-PPO (Pass\@16)                  | 30.00   | 72.29  | 86.80   | 30.15   | 46.96    | 53.24 |
>
> The results of Pass@8/16 accuracy (%) on math reasoning benchmarks of 7B models:
> | Model                               | AIME'24 | AMC'23 | MATH500 | Minerva | Olympiad | Avg.  |
> | ----------------------------------- | ------- | ------ | ------- | ------- | -------- | ----- |
> | R1-Distill-Qwen-7B (Base, Pass\@8)  | 63.33   | 83.13  | 93.80   | 47.79   | 55.11    | 68.63 |
> | ReMix-PPO (Pass\@8)                 | 76.67   | 92.77  | 95.80   | 50.74   | 63.70    | 75.94 |
> | R1-Distill-Qwen-7B (Base, Pass\@16) | 73.33   | 85.54  | 94.80   | 51.10   | 57.19    | 72.39 |
> | ReMix-PPO (Pass\@16)                | 80.00   | 92.77  | 95.40   | 53.31   | 65.33    | 77.36 |
>
> ___
>
> > **"Vanilla GRPO has a length normalization term, while some variants like Dr. GRPO [3] removed it. Did you apply length normalization in your experiments?"**
>
> Our main results use verl’s[1] default aggregation, token-mean loss, defined as a single global mean over log-probabilities of all tokens in the batch (flatten tokens across the batch and average once).
>
> For the GRPO variants mentioned by the reviewer, we add a series of ReMix-Dr.GRPO experiments. Follow Dr.GRPO, we compute the loss as a per-sequence token-sum followed by a batch mean (i.e., no length normalization), and we disable standard-deviation scaling in advantage normalization (no division by std). We set the max response length = 4096 (for swift experiments due to the stringent schedule of ICLR discussion stage) and evaluate the models with temperature=0.7, AIME/AMC with pass@1 (avg@32) and others with pass@1. All other experimental settings follow those in the main text.
>
> As shown in the table below, ReMix-Dr.GRPO delivers consistent improvements over Dr.GRPO at limited cost, mirroring the finding in the main body of our paper (ReMix-GRPO ≥ GRPO). Because ReMix is a plug-in layer for on-policy RFT methods, with safely reusing historical rollouts to improve sample efficiency and preserving training stability. Consequently, even when the backbone algorithm is switched from GRPO to Dr.GRPO, ReMix remains a suitable and effective drop-in.
>
> | Model | AIME24 | AMC23 | MATH–500 | Minerva | Olympiad | Avg. | Cost |
> | :--- | :--- | :--- | :--- | :--- | :--- | :--- | :--- |
> | R1–distilled–Qwen–1.5B(Base Model) | 12.50 | 40.96 | 68.20 | 19.49 | 27.70 | 33.77 | N/A |
> | Dr.GRPO (350 steps) | 21.25 | 63.10 | 80.10 | 27.21 | 41.33 | 46.60 | 0.358M |
> | Dr.GRPO (400 steps) | 26.25 | 59.94 | 80.60 | 26.57 | 41.03 | 46.86 | 0.409M |
> | **ReMix–Dr.GRPO (325 steps)** | **22.50** | **63.25** | **81.60** | **26.47** | **41.63** | **47.09** | **0.291M** |
> | **ReMix–Dr.GRPO (400 steps)** | **28.75** | **62.65** | **82.00** | **25.74** | **42.96** | **48.42** | **0.368M** |
>
> [1] https://github.com/volcengine/verl

---

> ### Author Response · Authors · 2025-11-25
> **Response to Reviewer h1SL (2/5)**
>
> > **"Only trained on Qwen model", Additional Results for Llama-8B Base Model**
>
> We appreciate the reviewer's constructive comment. Following common practice in RFT for LLMs reasoning [2, 3, 4] , our main experiments use the Qwen series. Nonetheless, we agree that conducting experiments on another base model familiy is significant. Thus, we add a cross-family evaluation on r1-distilled-Llama-8B to expand the scope of our evaluation and demonstrate that our method is not tied to a specific model family and is generally applicable. We evaluate the models with temperature=0.7, AIME/AMC(small datasets) with pass@1 (avg@32) and others with pass@1, all other experimental settings strictly follow the main text.
>
> We reports the best checkpoint within the training process of PPO and ReMix-PPO upon r1-distilled-Llama-8B (which are preliminary results as the experiments are still in progress) in the table below. These results  shows that ReMix-PPO effectively improves the performance over r1-distilled-Llama-8B and outperforms PPO in an efficient manner as well.
>
> This preliminarily indicate that our method is effective for Llama-based LLMs and echoes our claim that ReMix can be plugged into standard on-policy RFT to improve efficiency without sacrificing final performance.
>
> | Model | AIME24 | AMC23 | MATH–500 | Minerva | Olympiad | Avg. | Cost |
> | :--- | :--- | :--- | :--- | :--- | :--- | :--- | :--- |
> | R1–distilled–Llama–8B (Base Model) | 26.46 | 62.69 | 82.00 | 25.37 | 42.67 | 47.84 | N/A |
> | PPO (50 steps) | 34.58 | 70.63 | 85.40 | 26.10 | 45.18 | 52.38 | 0.013M |
> | **ReMix–PPO (25 steps)** | **39.48** | **75.04** | **88.00** | **28.68** | **48.74** | **55.99** | **0.004M** |
>
> For the experiments on r1-distilled-Llama-8B base, we are still running more and longer experiments. We will add the additional results in our revision to strengthen our experimental evaluation.
>
> [2] GPG: A Simple and Strong Reinforcement Learning Baseline for Model Reasoning. arXiv preprint arXiv: 2504.02546, 2025.
>
> [3] Fastcurl: Curriculum reinforcement learning with progressive context extension for efficient training r1-like reasoning models. arXiv preprint arXiv:2503.17287, 2025.
>
> [4] Deepscaler: Surpassing o1-preview with a 1.5b model by scaling rl, 2025.
>
> ___
>
> > **"Why does ReMix-GRPO make the model even worse then base model for AIME in table 1?"**
>
> We traced this observation to the evaluation protocol: the results in Table 1 are the pass@1 scores with greedy decoding, which works for larger test sets but can be high-variance on small test sets like AIME/AMC.
>
> To remedy this and ensure fairness, we evaluated ReMix-GRPO AIME24 and AMC23 with pass@1 (avg@32) under the same rollout and stochastic decoding settings as Appendix I (where we evaluate ReMix-PPO with avg@32 against different baselines). The table below reports Base vs. ReMix-PPO vs. ReMix-GRPO at 1.5B and 7B.
>
> **When evaluated in terms of Pass@1 (Avg@32), ReMix shows consistent and clear gains, which demonstrates the effectiveness on AIME’24 and AMC’23.**
>
> | Model | AIME24 | AMC23 |
> | :--- | :--- | :--- |
> | R1–distilled–Qwen–1.5B (Base Model) | 17.92 | 44.58 |
> | **ReMix–GRPO–1.5B (75 steps)** | **25.10** | **60.17** |
> | **ReMix–PPO–1.5B (350 steps)** | **29.08** | **64.04** |
> | R1–distilled–Qwen–7B (Base Model) | 37.53 | 66.55 |
> | **ReMix–GRPO–7B (200 steps)** | **49.48** | **82.12** |
> | **ReMix–PPO–7B (75 steps)** | **50.31** | **77.78** |
>
> We appreciate the reviewer for pointing out this and we will add these results in our revision.

---

> ### Author Response · Authors · 2025-11-25
> **Response to Reviewer h1SL (3/5)**
>
> > The Choices of Policy Reincarnation Trigger Step $T$
>
> In our pipeline, _$T$_ is the hand-over step that switches from the off-policy Mix-PPG phase to the on-policy phase with reference model changed too. We use it to harvest early data-efficiency from replay historical data and then let on-policy optimization continue improving stably.
> To address the reviewer’s concern about how the trigger step _$T$_ was chosen and how sensitive performance is to this choice, we provide a controlled sweep _$T \in \{25, 50, 100, 200\}$_ with 500-step runs, whose experimental settings are all the same as those of the main text. The results are shown in the table below. **For _$\(T \neq 25\)$_, we report two checkpoints per run, the first time the macro average reaches ≳49 (“early lift”) and the final plateau.**
>
> | Model | AIME24 | AMC23 | MATH–500 | Minerva | Olympiad | Avg. |
> | :--- | :--- | :--- | :--- | :--- | :--- | :--- |
> | T=25 (at 350 steps) | 33.33 | 65.06 | 81.20 | 27.57 | 39.26 | 49.29 |
> | T=50 (at 75 steps) | 43.33 | 63.86 | 79.60 | 26.84 | 39.41 | 50.61 |
> | T=50 (at 500 steps) | 40.00 | 63.86 | 83.00 | 26.84 | 43.70 | 51.52 |
> | T=100 (at 225 steps) | 30.00 | 69.88 | 81.80 | 24.63 | 41.78 | 49.62 |
> | T=100 (at 325 steps) | 36.67 | 69.88 | 82.00 | 30.15 | 41.78 | 52.09 |
> | T=200 (at 100 steps) | 30.00 | 66.27 | 81.00 | 29.41 | 43.26 | 49.99 |
> | T=200 (at 500 steps) | 16.67 | 49.40 | 77.00 | 16.67 | 38.67 | 40.76 |
>
> The pattern is consistent: a very small _$T$_ (25) slows down performance improvement; a very large _$T$_ (200) lifts early but later stalls due to amplified off-policyness; **a moderate _$T$_ (50–100) yields both a strong early lift and the best final averages. We also find this choice to broadly work well across our experimental cases, achieving a favorable efficiency–performance trade-off.**
>
> To take a step further, we believe that it is also possible to propose an adaptive strategy for the dynamic choice of the trigger step _$T$_. For example, we can use the reflection rate or some measurements of off-policyness as a reference, and set a threshold to trigger policy reincarnation. More sophisticated strategy can aslo be designed. This is beyond the scope of this paper and thus we consider it as a future direction.
>
> ___
>
> >  **The Specific Form of the Decay Schedule for KL-convex Constraint**
>
> In our experiments, we found using **a slowly decaying KL-Convex coefficient consistently beat a fixed coefficient under matched budgets.  And we also tried other forms of decay schedule, (e.g., exponential decay) and did not found a significant difference.** Therefore, we stick to the simple linear decay schedule, to avoid over-sophisticated design as widely used in prior works.
> We appreciate the reviewer for point this out. We will add the clarification on this point and related experimental results in our revision.

---

> ### Author Response · Authors · 2025-11-26
> **Response to Reviewer h1SL (4/5)**
>
> > **What is the exact formulation of distribution $i \sim \nu$ used in practice, and is there any theoretical explanation?**
>
> _$\nu$_ is the distribution of policy index $i$, where $i \in \{0, 1, \dots, N\}$. The theoretical explanation for this should trace back to how Trust-Region Policy Optimization (TRPO)[5] approximates the trust region.
>
> For $i = 0$, i.e., the standard PPO (on-policy case), the trust region $\alpha^2 = [ \max_s TV(\pi_k, \pi)(s) ]^2$ or its upper bound $D_{\text{KL}}(\pi_k, \pi)$ (this is can be found in Theorem 1, Eq.8 in TRPO paper) is approximated by using the expectation regarding $d^{\pi_k}$ to replace the maximum case. In turn, the approximated trust region is $\tilde{\alpha}^2 = [ \mathbb{E}_{s \sim d^{\pi_k}} TV(\pi_k, \pi)(s) ]^2$.
>
> In practices of PPO and TRPO, this approximation works well in many problems. Intuitively, this is because the new policy $\pi$ (post-update) should not differ a lot with the current policy $\pi_k$ (prior-update), the distribution $ d^{\pi_k} $ works as an effective surrogate. When it moves on to the off-policy case where $i > 0$ [7], the approximation for the trust region could be no longer effective for stable policy optimization due to the increasing discrepancy between $ d^{\pi} $ and $ d^{\pi_{k-i}} $ as the increase of $i$ (i.e., for older historical policies).
>
> In our work, we empirically found that using too old histrical policies can introduce large off-policyness which makes the training unstable. Our empirical observation aligns with our discussion on the theoretical explanation of the approximation of trust region above. Therefore, in practice, we use $N = 2$, $p=0.4$ and uniform distribution for off-policy data. **This is equivalent to the policy index distribution $\nu = [0.6, 0.2, 0.2]$. We believe that one important future direction is to study how to replay off-policy data better (e.g., taking an analogy to Priority Experience Replay, etc.).**
>
> We appreciate the reviewer for pointing this inspiring question. We will add these discussions in our revision.
>
> ___
>
> > **No theoretical explanation for the design of KLC loss and ReMix algorithm.**
>
> **The KLC loss is theoretically grounded in Conservative Policy Iteration (CPI) from prior work. The CPI paper[6] proves that iteratively refining the reference policy guarantees monotonic improvement** (Proposition 1: $ V^{\bar{\pi}^{\ast} } (s) \geq V^{\bar{\pi}} (s) $) and support preservation ( $\bar \pi^{\ast} (a|s)=0$ wherever $\bar{\pi}(a|s)=0$ ). Our convex combination $L_{\mathrm{KLC}} = \lambda D_{\mathrm{KL}}(\pi_{\theta}||\pi_{\mathrm{base}}) + (1-\lambda) D_{\mathrm{KL}}(\pi_{\theta}||\pi_{k-1})$ directly implements this—where the $\pi_{k-1}$ term acts as CPI's dynamic reference policy to prevent OOD queries, while the $\pi_{\mathrm{base}}$ term extends the framework to preserve foundational capabilities. Furthermore, CPI's Theorem 1 identifies that multi-step actor-critic implementations suffer from high variance, and the authors explicitly recommend adding behavior regularization (their Eq. 6) to constrain policies to data support. Our decaying $\lambda(t)$ schedule dynamically balances this trade-off, shifting from conservative exploration to aggressive refinement. Thus, KLC inherits CPI's theoretical guarantees of improvement and convergence while addressing practical stability challenges in iterative LLM fine-tuning.
>
> ReMix's design is theoretically supported by established frameworks: **Mix-PPG extends proximal policy gradients to off-policy data reuse [7]**, but its Whipping Effect, where longer responses amplify importance sampling ratios, causing bias toward short answers and self-reflection collapse, motivates the other components. **The KLC loss adapts Conservative Policy Iteration [6, 8]:** its convex combination of KL terms inherits CPI's guarantees of monotonic improvement and support preservation (preventing OOD queries), while the decaying $\lambda (t)$ schedule implements a curriculum shifting from stability to flexibility. **Policy Reincarnation[9] provides the two-stage framework:** Mix-PPG for rapid early gains, then switching to on-policy PPO with $\pi_{T}$ as the new reference, to achieve stable convergence without asymptotic compromise.
>
> [5] Trust Region Policy Optimization. In ICML, 2015.
>
> [6] Iteratively refined behavior regularization for offline reinforcement learning. NeurIPS, 37:56215–56243, 2024.
>
> [7] Generalized proximal policy optimization with sample reuse. In NeurIPS, pages 11909–11919, 2021.
>
> [8] Politex: Regret bounds for policy iteration using expert prediction. In ICML, 2019.
>
> [9] Reincarnating reinforcement learning: Reusing prior computation to accelerate progress. In NeurIPS, 2022.

---

> ### Author Response · Authors · 2025-11-26
> **Response to Reviewer h1SL (5/5)**
>
> > **The Missing Derivation for Eq.6**
>
> We appreicate the reviewer for pointing out this. As we mentioned in our paper,  we conduct the formal analysis for Mix-PPG by using a similar form of average loss as in [10].
>
> Here, we provide the complete derivation below. Starting from Eq.3, that is the definition of loss function for Mix-PPG, we re-write the loss function from the original per-sample expectation (i.e., $(s,a) \sim d^{\pi\_{k-i}}$) form to the per-trajectory expectation form (i.e., $\tau \sim d^{\pi_{k-i}}$). This derives:
>
> $$
> L^{\text{Mix-PPG}}_{\text{Avg}} (\theta) = - \mathbb{E}\_{i \sim \nu} \bigg[ \mathbb{E}\_{\tau \sim d^{\pi\_{k-i}}} \frac{1}{H\_{\tau}} \sum\_{h=0}^{H\_{\tau}} \min \Big( r^{k-i}\_{\theta}(s\_h, a\_h) A^{\pi\_k}(s\_h,a\_h), \\
> \text{clip} \left( r^{k-i}\_{\theta}(s\_h,a\_h), \frac{\pi\_k(a\_h \mid s\_h)}{\pi\_{k-i}(a\_h \mid s\_h)} - \epsilon, \frac{\pi\_k(a\_h \mid s\_h)}{\pi\_{k-i}(a\_h \mid s\_h)} + \epsilon \right) A^{\pi\_k}(s\_h,a\_h) \Big) \bigg],
> $$
>
> where $H_{\tau}$ is response length for trajectory $\tau$.
>
> By **dropping the clipping range (i.e., ignoring the out-of-clipping range cases which have no gradient) and simplifying the expression** by omitting subscript notations, expectation notations, etc., we only keep the proportional relationship for the analysis.
> **This then leads to: $L^{\text{Mix-PPG}}\_{\text{Avg}} = - \frac{1}{H} \sum\_{h=0}^{H}  r^{k-i}\_{\theta} A^{\pi\_k}$, which is the exact the form we used in Eq.6.**
>
> We will include the detailed derivation along with more clarification in our revision.
>
> [10] Concise reasoning via reinforcement learning. arXiv preprint arXiv:2504.05185, 2025.
>
> ___
>
> > **On the Information Conveyed by Figure 4**
>
> Figure 4 is mainly intended to diagnose the effect of Mix-PPG and Mix-PPG with an increased UTD ratio on reasoning behavior: both of them suffer from uncontrolled off-policyness, leading to clear pathologies in the later learning stage --- policy degradation and collapse, respectively.
>
> In particular, the three plots on the right side (i.e., response length, relative response length and self-reflection) together illustrates a clear pattern: as responses become shorter under aggressive data reuse, the model’s self-reflection behavior also drops, which in turn hurts final performance and can even lead to collapse. For example, Mix-PPG with an increased UTD ratio drives response length down very quickly, causing self-reflection to fall sharply and ultimately leading to a performance collapse. In contrast, PPO and ReMix-PPO do not exhibit this shortening-then-collapse pattern.
>
> For the pass@1 curves in Figure 4, the overall pattern is consistent with our analysis: Mix-PPG with an increased UTD ratio rises quickly at first and then stops improving or even drops, while PPO and ReMix-PPO remain stable. **For the PPO vs. ReMix-PPO comparison specifically, the results of same test set in Figure 8 (and more in Figure 3, 5) show the trend much more clearly** without the visual distraction of the curves for Mix-PPG and Mix-PPG with an increased UTD ratio: ReMix-PPO improves faster in the early stage and achieves better final performance. These trends match the behaviors discussed in Section 4.4.

---

> ### Author Response · Authors · 2025-11-26
>
> Dear Reviewer h1SL,
>
> We sincerely appreciate your valuable time and constructive comments. As the end of the discussion stage is approaching, we would greatly appreciate it if you could confirm whether our response has addressed your concerns and re-evaluate our work accordingly.
>
> We hope that our clarifications and additional experimental results have addressed all your concerns. We are also more than willing to answer any further questions during the discussion stage.

---

> ### Comment · Reviewer_h1SL · 2025-11-26
>
> Thank you for your response. Most of my concerns have resolved. It is interesting that ReMix–PPO on 7B model obtains minimal improvement (or even worse) in pass@8/16 on Minerva, which doesn't happen for 1.5B model. Is there any explanation?

---

> > ### Comment · Reviewer_h1SL · 2025-11-26
> >
> > Raised my rating to 8.

---

> > > ### Author Response · Authors · 2025-12-03
> > >
> > > Thanks for the follow-up question.
> > >
> > > Regarding the performance on the 7B model, we investigated the issue and found that the discrepancy was caused by inconsistent rollout settings. Specifically, we initially adapted the rollout configurations (e.g., parallelism) to accommodate different hardware constraints, which inadvertently impacted the sampling randomness and diversity.
> > >
> > > We have since aligned the evaluation settings for the 7B model. The re-evaluation confirms that ReMix-PPO consistently outperforms the base model on Minerva. Specifically, ReMix-PPO achieves 50.74% (pass@8) and 53.31% (pass@16), compared to 47.79% and 51.10% for the base model. We have updated the table in the reply "Response to Reviewer h1SL (1/5)" to reflect these changes: while the 1.5B results remain unchanged, the 7B results have been corrected. These updates will also be included in the final revision.

---

### Official Review · Reviewer_pHkv · 2025-10-28

**Soundness:** 3
**Presentation:** 3
**Contribution:** 3
**Rating:** 6
**Confidence:** 5

**Summary:**

This paper presents **ReMix** (*Reincarnating Mix-policy Proximal Policy Optimization*), a framework designed to make reinforcement fine-tuning (RFT) of large language models more sample-efficient by effectively leveraging off-policy data.

ReMix extends standard on-policy approaches such as **PPO** and **GRPO** through three main components:

- **Mix-policy proximal gradient:** Combines on- and off-policy trajectories to achieve higher update-to-data (UTD) ratios and more efficient data reuse.
- **KL-convex constraint:** Interpolates between the base and current policies to ensure updates remain both stable and adaptive.
- **Policy reincarnation:** Periodically resets the base model to the current policy, enabling a smooth transition from rapid early learning to stable convergence.

Across five math-reasoning benchmarks—**AIME24**, **AMC23**, **Minerva**, **OlympiadBench**, and **MATH500**—ReMix achieves state-of-the-art Pass@1 accuracy while using **30×–450× fewer rollouts** than leading baselines such as **DeepScaleR** and **AceReason**.
The authors also analyze the behavioral effects of off-policy learning, observing patterns like shorter responses and reduced self-reflection when off-policyness becomes excessive.

**Strengths:**

**Clear Motivation & Contribution** – Addresses the well-known issue of *sample inefficiency* in RFT with a principled and well-structured solution.
**Novel Integration of Off-Policy Concepts** – The combination of mix-policy updates, convex KL constraint, and policy reincarnation is original and intuitively justified.
**Strong Empirical Results** – Achieves SOTA reasoning accuracy while reducing training cost by 30×–450× across multiple benchmarks.
**Comprehensive Experiments** – Includes ablation studies, efficiency–accuracy trade-off curves, and behavioral analyses (e.g., response length, self-reflection rate).
**Readable and Well-Presented** – The paper is clearly written and easy to follow.

**Weaknesses:**

Overall, there are no major weaknesses in this paper. A minor concern is that the authors did not compare **ReMix** with other *off-policy* algorithms, relying only on *on-policy* baselines. Including comparisons with off-policy methods in future work would provide a more comprehensive evaluation and strengthen the empirical claims.

**Questions:**

1. In Eq. (3), why do the authors use \(\frac{\pi_K}{\pi_{K-i}}\) to control the trust region? What is the motivation or benefit compared to using a fixed constant region? It would be helpful to see empirical evidence or ablation results demonstrating that this adaptive trust-region design improves performance.
2. In Fig. 3, ReMix-PPO appears to achieve roughly \(6\times\) faster improvement than vanilla PPO in the early training stage. Does this imply that early trajectories are reused multiple times for updates? If so, could this repeated use of the same trajectories lead to training instability or bias accumulation?

---

> ### Author Response · Authors · 2025-11-25
> **Response to Reviewer pHkv (1/2)**
>
> We sincerely appreciate the reviewer's very careful and constructive comments. We are glad to see that the reviewer recognizes our contributions. Our response below aims to address the questions raised by the reviewer and discuss the constructive comments.
>
> We look forward to addressing any additional concerns and engaging in further productive discussions during the discussion phase.
>
> ___
>
> > **"In Eq. (3), why do the authors use ($\frac{\pi_k}{\pi_{k-i}}$) to control the trust region? What is the motivation or benefit compared to using a fixed constant region? It would be helpful to see empirical evidence or ablation results demonstrating that this adaptive trust-region design."**
>
> Different from the standard PPO, which has a clipping range $(1-\epsilon, 1+\epsilon)$, the generalized clipping range for Generalized Proximal Policy Gradient is $\big( \frac{\pi_k}{\pi_{k-i}} -\epsilon, \frac{\pi_k}{\pi_{k-i}} +\epsilon \big)$.
>
> The source of this distinction is the incoporation of off-policy data distribution $(s,a) \sim d^{\pi_{k-i}}$ (recall that $\pi_k$ is the current policy to update). To be more concrete, the clipping mechanism can be rewritten to prevent $\big| \frac{\pi}{\pi_k} - 1 \big|$ from exceeding $\epsilon$. In principle, this is equivallent to constraining the total variance between $\pi$ and $\pi_k$ , which is very close to the original principle of TRPO. Formally, there is $\mathbb{E}\_{s\sim d^{\pi\_k} } [ TV(\pi, \pi\_k)(s) ] = \frac{1}{2} \mathbb{E}\_{(s,a) \sim d^{\pi\_k} } \big[ \big| \frac{\pi}{\pi\_k} - 1 \big| \big]$, where a proof can be found in Appendix A.2 of [1].
>
> When off-policy data distribution ($(s,a) \sim d^{\pi_{k-i}}$) is used, we can have a new form of the total variance between $\pi$ and $\pi_k$ regarding off-policy data distribution as: $\mathbb{E}\_{i \sim \nu} [ \mathbb{E}\_{s\sim d^{\pi\_{k-i}} } [ TV(\pi, \pi\_k)(s) ] ] = \frac{1}{2} \mathbb{E}\_{i \sim \nu} [ \mathbb{E}\_{(s,a) \sim d^{\pi\_{k-i}} } \big[ \big| \frac{\pi}{\pi\_{k-i}} - \frac{\pi\_k}{\pi\_{k-i}} \big| \big] ]$, where a proof can be found in Appendix A.3 of [1] . This further derives the the clipping range $\big( \frac{\pi_k}{\pi_{k-i}} -\epsilon, \frac{\pi_k}{\pi_{k-i}} +\epsilon \big)$.
>
> Intuitively, the clipping center changes from 1 to $\frac{\pi_k}{\pi_{k-i}}$ and one may notice that when $i=0$ (i.e., the on-policy case), $\big( \frac{\pi_k}{\pi_{k-i}} -\epsilon, \frac{\pi_k}{\pi_{k-i}} +\epsilon \big)$ becomes the standard form. This is because when constraining the trust region (i.e., the total variance between $\pi$ and $\pi_k$) regarding an off-policy data distribution, the existence of the importance sampling of $\pi_{k-1}$ also shifts the clipping center.
>
> In short, $\frac{\pi_k}{\pi_{k-i}}$ **is a principled choice of clipping center which is derived from the generalization of proximal policy gradient from on-policy data distribution to off-policy data distribution.** Using the standard form for off-policy data distribution ("fixed constant region") should in principle introduce a bias on the trust region that constrains policy learning.
>
> [1] Generalized proximal policy optimization with sample reuse. In NeurIPS, pp. 11909–11919, 2021.
>
> ___
>
> > **On the Suggestion of Including Comparisons with Off-policy Methods in Future Work**
>
> We appreciate the suggestion and agree broader comparisons with off-policy methods are valuable and necessary. As we mentioned in Line 295, Section 4.1 and further explained in Appendix F.3, we are aware of there are a few related off-policy methods for LLM RFT, however, they are not directly comparable for different reasons.
>
> To be specific, for RePO, their models are trained with a 1024-token maximum response length, which underperforms on long-form mathematical reasoning. LUFFY relies on guidance from a much stronger external model (e.g., DeepSeek-R1), a teacher-guided setting, which is orthogonal to our teacher-free setup. And for SPO, AGRO, AsymRE, and Tapered Off-policy REINFORCE, there are no public checkpoints or stable, size-matched implementations at the 1.5B and 7B scales we study --- retraining bespoke replicas would require substantial method-specific tuning and compute, confounding a fixed-budget comparison and reducing reproducibility.
>
> For future works, we will open-source our current training/evaluation code to facilitate reproducibility and community reuse, and include off-policy methods once size-matched, stable checkpoints are available.

---

> ### Author Response · Authors · 2025-11-25
> **Response to Reviewer pHkv (2/2)**
>
> > **"In Fig. 3, ReMix-PPO appears to achieve roughly (6\times) faster improvement than vanilla PPO in the early training stage. Does this imply that early trajectories are reused multiple times for updates? If so, could this repeated use of the same trajectories lead to training instability or bias accumulation?"**
>
> In the context of our work, "early trajectories are reused multiple times for updates" can be understood in two ways: off-policy training and using a UTD ratio $m \ge 2$.
>
> For off-policy training, early trajectories are reused multiple times for updates as for each batch training, we will sample from both online trajectories and from historical trajectories (i.e., off-policy data), which is controlled by the historical policy window size $N$ and off-policy data portion $p$ (see Algorithm 1). This is indeed what off-policy training or off-policy RL typically means. For using a UTD ratio $m \ge 2$, each batch (which is a mix of on-policy trajectories and off-policy trajectories as mentioned above) will be used for policy update for consecutive $m$ training steps. This design choice is inspired by REDQ[2], which shows that taking multiple gradient updates per data point can substantially improve data efficiency.
>
> Both off-policy training and using a UTD ratio $m \ge 2$ will increase the off-policyness, making the training process more prone to off-policy bias (accumulation) and instability (or even collapse). This is also shown in Figure 4 and highlighted by Takeaway 3.
> Although controlling off-policy bias has been studied in classic RL, how to control it effectively in the context of LLM reasoning is still under-explored. The major contributions of our work lie on this point. **In ReMix, off-policyness is harnessed by using Generalized Proximal Policy Gradient objective as well as a proper off-policy data portion $p$; while stability is largely ensured by the policy reincarnation at step $T$ (which means stopping replay reuse before bias accumulation may cause a collapse of training).**
>
> As In Figure 4 of our paper, "Mix-PPG" and  "Mix-PPG with an Increased UTD Ratio" show that a later-stage degradation and a policy collapse respectively, which we deem is caused by the accumulation of off-policy bias. **In contrast, the curve of ReMix-PPO shows that our method effectively balance the early-stage efficiency and fianl performance.**
>
> [2] Randomized ensembled double q-learning: Learning fast without a model. In ICLR, 2021.

---

> ### Author Response · Authors · 2025-11-26
>
> Dear Reviewer pHkv,
>
> We sincerely appreciate your valuable time and constructive comments. As the end of the discussion stage is approaching, we would greatly appreciate it if you could confirm whether our response has addressed your concerns and re-evaluate our work accordingly.
>
> We hope that our clarifications have addressed all your concerns. We are also more than willing to answer any further questions during the discussion stage.

---

### Official Review · Reviewer_cESh · 2025-10-31

**Soundness:** 2
**Presentation:** 3
**Contribution:** 2
**Rating:** 6
**Confidence:** 2

**Summary:**

This paper proposes ReMix, a method for efficient off-policy Reinforcement Finetuning (RFT) of Large Language Models. ReMix integrates three key components: a Mix-Policy Proximal Policy Gradient to utilize historical data with an increased Update-to-Data ratio, a KL-Convex policy constraint that dynamically blends penalties from the base model and the previous policy, and a Policy Reincarnation mechanism that switches the base model and reverts to on-policy training mid-way. The authors demonstrate strong empirical results on mathematical reasoning benchmarks, showing that ReMix-based models can achieve state-of-the-art or competitive performance with a dramatic reduction (30x to 450x) in the amount of rollout data required compared to several strong baselines.

**Strengths:**

- Well-Designed and Synergistic Method: The three-component architecture is intuitive and addresses key challenges in off-policy RFT. The components work synergistically, as shown by the ablations, with Policy Reincarnation being a particularly clever and novel idea for bridging early-stage efficiency and late-stage stability.

- Rigorous Ablation Studies: The ablation study in Table 3 is comprehensive and effectively validates the contribution of each proposed component, clearly showing that the full combination is necessary for optimal performance.

**Weaknesses:**

- Overly Optimistic and Potentially Misleading Framing: The paper's central claim of "30x to 450x" reduction in rollout data is framed in a way that overstates the advantage. This is achieved by comparing ReMix's early-stage performance against baselines' final performance (e.g., ReMix-PPO at 100 steps vs. PPO at 900 steps). A comparison of final performances shows a much more modest efficiency gain, undermining the dramatic narrative.
- Limited Scope of Evaluation: The empirical validation is constrained in two key dimensions: model scale (≤7B parameters) and task domain (mathematical reasoning only). This leaves the generality of ReMix an open question. Its effectiveness on larger models (e.g, 70B+) or on fundamentally different tasks (e.g., code generation) is not established. Furthermore, the absence of ReMix-GRPO results for the 7B model is a notable omission, as it prevents a consistent assessment of the method's performance across different underlying RL algorithms at a more capable scale.

**Questions:**

- The KL-Convex constraint uses a dynamically decaying coefficient $\lambda(t)$ to balance between the base and previous policy. The design of this decay schedule $\lambda(t) = \max\left(1 − 0.1 \cdot \lceil \max(t − 50, 0) / 10 \rceil, 0.5\right)$ appears to be heuristic. Could you provide an analysis that justifies this specific form?
- The Policy Reincarnation mechanism is a critical component, yet its triggering is based on a predetermined step $T$. How was this specific hyperparameter chosen, and what is the sensitivity of the final performance to this choice?
- The analysis reveals a strong "preference for shorter responses." Do you think this bias could be harmful in tasks where comprehensive reasoning is crucial, and if so, how could ReMix be adapted to control for this effect?

---

> ### Author Response · Authors · 2025-11-25
> **Response to Reviewer cESh (1/3)**
>
> We sincerely appreciate the reviewer's very careful and constructive comments. We are glad to see that the reviewer recognizes our method design and ablation study. The concerns of the reviewer focus on the scope of experiments, the choices of hyperparameters, etc. Our response below aims to address these important points one by one with additional experimental results and clarification.
> We sincerely hope that the reviewer could re-evaluate our work according to our response and additional results. We also look forward to addressing any additional concerns and engaging in further productive discussions during the discussion phase.
> ___
>
> > **Beyond Math Reasoning, Additional Results for ReMix in Code Generation Domain**
>
> We appreciate that the reviewer points this out. To address this, we conduct additional experiments for code-generation at 1.5B and 7B scales. For training, we follow the common practice in code-RFT and use the **Skywork-OR1-RL-Data dataset**. Because coding prompts are substantially longer than math tasks, we set max_prompt_length = 1600. We evaluate our models on **LiveCodeBench(8/1/24-2/1/25)** with temperature=0.7. All other experimental settings follow those in the main text.
>
> The additional experimental results for code generation are presented in the tables below. **Across both scales, ReMix-PPO improves pass@1 over PPO while using much less cost.** These gains in a fundamentally different domain demonstrate that our method is not specific to math-reasoning and remains effective for code generation, where outputs are typically longer and structurally constrained.
>
> | Model | LiveCodeBench (8/1/24–2/1/25) | Cost |
> | :--- | :--- | :--- |
> | R1–distilled–Qwen–1.5B(Base Model) | 12.90 | N/A |
> | PPO(375steps) | 19.35 | 0.096M |
> | **ReMix–PPO(300steps)** | **20.07** | **0.066M** |
> | **ReMix–PPO(475steps)** | **24.01** | **0.111M** |
>
> | Model | LiveCodeBench (8/1/24–2/1/25) | Cost |
> | :--- | :--- | :--- |
> | R1–distilled–Qwen–7B(Base Model) | 26.16 | N/A |
> | PPO(200steps) | 33.33 | 0.051M |
> | **ReMix–PPO(100steps)** | **34.41** | **0.015M** |
>
> We will add these additional results in our revision.
>
> ___
>
> > **The Results for ReMix-GRPO at 7B scale**
>
> We appreciate the reviewer for pointing this out. We now include the previously missing 7B ReMix-GRPO.
> As shown in the table below, **it matches ReMix-PPO’s best performance and outperforms the strongest baseline (AceReason-Nemotron) while having markedly less cost.** All experimental settings follow those in the main text. We will include these results in our revision.
>
> | Model | AIME24 | AMC23 | MATH–500 | Minerva | Olympiad | Avg. | Cost |
> | :--- | :--- | :--- | :--- | :--- | :--- | :--- | :--- |
> | R1–distilled–Qwen–7B(Base Model) | 33.33 | 68.68 | 83.80 | 30.15 | 44.44 | 52.08 | N/A |
> | AceReason–Nemotron | 60.00 | 80.72 | 89.00 | 36.40 | 50.07 | 63.24 | > 3.584M |
> | **ReMix–GRPO(75steps)** | **53.33** | **81.93** | **91.60** | **39.34** | **53.19** | **63.88** | **0.046M** |
> | **ReMix–GRPO(200steps)** | **56.67** | **80.72** | **90.60** | **40.07** | **53.78** | **64.37** | **0.163M** |
> | **ReMix–PPO(75steps)** | **63.33** | **78.31** | **90.20** | **37.50** | **52.59** | **64.39** | **0.011M** |

---

> ### Author Response · Authors · 2025-11-25
> **Response to Reviewer cESh (2/3)**
>
> > **"The paper's central claim of "30x to 450x" reduction in rollout data is framed in a way that overstates the advantage. This is achieved by comparing ReMix's early-stage performance against baselines' final performance (e.g., ReMix-PPO at 100 steps vs. PPO at 900 steps). A comparison of final performances shows a much more modest efficiency gain, undermining the dramatic narrative"**
>
> **The number 30x and 450x are calculated by comparing DeepScaleR/AceReason-Nemotron with ReMix at 1.5B/7B regarding Rollout Data Volume.** Note that DeepScaleR and AceReason-Nemotron (publicly accessable checkpoints) are the best-performing baseline models in our experiments at 1.5B and 7B scales, respectively. (Note that ReMix-PPO/GRPO matches or outperforms the two best baselines while vanilla PPO/GRPO does not.) **This is concretely described between Line 317-322, Table 1,2,7,8 in our submission. Therefore, this is NOT acchieved by comparing ReMix's early-stage performance against baselines' final performance (e.g., ReMix-PPO at 100 steps vs. PPO at 900 steps)**, as the reviewer mentioned.
>
> **For the comparison between PPO and ReMix-PPO (also GRPO v.s., ReMix-GRPO), we use the best checkpoints obtained within a specific training step budget for the results reported in Table 1 and 2  (as mentioned in Line 285).** This is the model-selection rule practitioners actually use (eg. Open-RS[1]). We then compute the training cost reduction of ReMix when it outperforms the scores of PPO or GRPO roughly at the first time. To be specific, for 1.5B models, **ReMix-PPO reaches 50.61 (100 steps, 0.020M rollouts)**, that is **10× fewer than PPO (50.61, 900 steps, 0.230M; and note 49.56, 500 steps, 0.128M)**; for 7B models, **ReMix-PPO (58.49, 25 steps, 0.003M; 63.27, 50 steps, 0.007M)** achieves a **6× reduction** compared to **PPO (58.87, 100 steps, 0.026M; 60.34, 200 steps, 0.051M)**.
>
> More intuitive views about training efficiency can also be observed by referring to the training curves in Figure 3 and 5.
>
> In short, ReMix typically reaches the same-level performance of vanilla PPO/GRPO and other baseline models earlier, exhibiting a lower Cost. This is also precisely the sample-efficiency advantage we seek to achieve in practice.
>
> [1] Reinforcement learning for reasoning in small llms: What works and what doesn’t. arXiv preprint arXiv:2503.16219, 2025.
>
> ___
>
> > **On the Sensitivity and Choices of Policy Reincarnation Trigger Step _$T$_**
>
> In our pipeline, _$T$_ is the hand-over step that switches from the off-policy Mix-PPG phase to the on-policy phase with reference model changed too. We use it to harvest early data-efficiency from replay historical data and then let on-policy optimization continue improving stably.
> To address the reviewer’s concern about how the trigger step _$T$_ was chosen and how sensitive performance is to this choice, we provide a controlled sweep _$T \in \{25, 50, 100, 200\}$_ with 500-step runs, whose experimental settings are all the same as those of the main text. The results are shown in the table below. **For _$\(T \neq 25\)$_, we report two checkpoints per run, the first time the macro average reaches ≳49 (“early lift”) and the final plateau.**
>
> | Model | AIME24 | AMC23 | MATH–500 | Minerva | Olympiad | Avg. |
> | :--- | :--- | :--- | :--- | :--- | :--- | :--- |
> | T=25 (at 350 steps) | 33.33 | 65.06 | 81.20 | 27.57 | 39.26 | 49.29 |
> | T=50 (at 75 steps) | 43.33 | 63.86 | 79.60 | 26.84 | 39.41 | 50.61 |
> | T=50 (at 500 steps) | 40.00 | 63.86 | 83.00 | 26.84 | 43.70 | 51.52 |
> | T=100 (at 225 steps) | 30.00 | 69.88 | 81.80 | 24.63 | 41.78 | 49.62 |
> | T=100 (at 325 steps) | 36.67 | 69.88 | 82.00 | 30.15 | 41.78 | 52.09 |
> | T=200 (at 100 steps) | 30.00 | 66.27 | 81.00 | 29.41 | 43.26 | 49.99 |
> | T=200 (at 500 steps) | 16.67 | 49.40 | 77.00 | 16.67 | 38.67 | 40.76 |
>
> The pattern is consistent: a very small _$T$_ (25) slows down performance improvement; a very large _$T$_ (200) lifts early but later stalls due to amplified off-policyness; **a moderate _$T$_ (50–100) yields both a strong early lift and the best final averages. We also find this choice to broadly work well across our experimental cases, achieving a favorable efficiency–performance trade-off.**
>
> To take a step further, we believe that it is also possible to propose an adaptive strategy for the dynamic choice of the trigger step _$T$_. For example, we can use the reflection rate or some measurements of off-policyness as a reference, and set a threshold to trigger policy reincarnation. More sophisticated strategy can aslo be designed. This is beyond the scope of this paper and thus we consider it as a future direction.

---

> ### Author Response · Authors · 2025-11-25
> **Response to Reviewer cESh (3/3)**
>
> >  **The Specific Form of the Decay Schedule for KL-convex Constraint**
>
> In our experiments, we found using **a slowly decaying KL-Convex coefficient consistently beat a fixed coefficient under matched budgets.  And we also tried other forms of decay schedule, (e.g., exponential decay) and did not found a significant difference.** Therefore, we stick to the simple linear decay schedule, to avoid over-sophisticated design as widely used in prior works.
> We appreciate the reviewer for point this out. We will add the clarification on this point and related experimental results in our revision.
>
> ___
>
> > **The analysis reveals a strong "preference for shorter responses." Do you think this bias could be harmful in tasks where comprehensive reasoning is crucial, and if so, how could ReMix be adapted to control for this effect?**
>
> We appreciate the reviewer for this inspiring question. In our opinion, the implicit preference for shorter responses can be either beneficial or harmful in different settings. For example, this can be a good thing in some practical settings with a limited response length. And also, this can also cripple the exploration for longer, new reasoning trajectories.
>
> We view this as a manifestication of the classic "exploitation-and-exploration" dilemma, which is one of the most fundamental dilemma in RL. This also makes sense intuitively: off-policy learning aims to best exploit from historical data, thus largely leaning on the side of exploitation and suppressing exploration.
>
> This inspires us to seek for a more sophiticated method to better combine exploitation and exploration, as mirrored in the literature of classic RL. One possible way is to encourage exploration when doing online policy rollout for more diverse responses. Classic principles like curiosity-driven exploration, uncertainty-driven exploration, etc can be revisited in the context of LLM RFT, while the 80-20 entropy characteristic of LLM can also be leveraged for better exploration. Additionally, auxilliary mechanisms like injecting refection tokens during online rollout and so on are also possible.
>
> With more diverse responses collected by a good exploration policy, the potential of off-policy RFT methods (e.g., ReMix) can be leveraged more thoroughly. At that point, the "limitation" could be converted into an "advantage".
>
> We appreciate the reviewer's valuable question, which really inspired us of a potential future direction.
>
> ___
>
> > **The suggestion on conducting experiments with larger models (e.g., 70B+)**
>
> We appreciate the reviewer's suggestion for broader evidence. For conducting experiments with larger scales (e.g., _$\ge 70$_B), we do not have sufficient resource to do this at this point, although we agree that it is of great importance.
> Our focus of this work is efficient off-policy RFT regarding rollout and training budgets. In line with community practice, where most recent reasoning RFT studies [2, 3, 4] evaluate at the 1.5B and 7B scales to ensure comparability and reproducibility, our main experiments anchor on these scales.
>
> [2] GPG: A Simple and Strong Reinforcement Learning Baseline for Model Reasoning. arXiv preprint arXiv: 2504.02546, 2025.
>
> [3] Fastcurl: Curriculum reinforcement learning with progressive context extension for efficient training r1-like reasoning models. arXiv preprint arXiv:2503.17287, 2025.
>
> [4] Deepscaler: Surpassing o1-preview with a 1.5b model by scaling rl, 2025.

---

> ### Author Response · Authors · 2025-11-26
>
> Dear Reviewer cESh,
>
> We sincerely appreciate your valuable time and constructive comments. As the end of the discussion stage is approaching, we would greatly appreciate it if you could confirm whether our response has addressed your concerns and re-evaluate our work accordingly.
>
> We hope that our clarifications and additional experimental results have addressed all your concerns. We are also more than willing to answer any further questions during the discussion stage.

---

### Author Response · Authors · 2025-12-03
**Summary of Rebuttal and Global Response**

We sincerely appreciate the Area Chair and all reviewers for their valuable time, constructive comments, and careful evaluation of our work. A summary of the author-reviewer rebuttal process and its outcomes are provided below. We have also uploaded a revision that includes all the additional experiments and discussions suggested in the reviewers' comments (all the modifications are highlighted in blue in the revision and the corresponding places are marked below).

The rebuttal directly addressed all the concerns raised by the reviewers.
- Empirically, we **extended our evaluation to the code generation domain (Section 4.5 in the revision)**, **reported pass@8 and pass@16 results (Section 4.5)**, **included additional base RFT algorithms like Dr.GRPO (Section 4.5)** (and DAPO as well yet in progress), and **added Deepseek-R1-Distilled-Llama-8B as another base model (Appendix M)**. All these additional experiments demonstrates the effectiveness of our proposed method across various settings. We also provided **a sensitivity analysis of the Policy Reincarnation trigger step $T$ (Appendix D.5)**, demonstrating the robustness of the hyperparameter choice.
- Theoretically, we provided the missing derivation for Eq. 6 (Appendix D.4), clarified the exact formulation for the distribution $i \sim \nu$ (Appendix D.2). Furthermore, we added detailed justifications for the design of the KLC loss, alongside an analysis of the specific form of the decay schedule for the KL-convex constraint(Appendix D.5 and Section 3.2).
These additions resolve the conceptual and empirical questions raised during the review.

The initial ratings were 6, 6, 6, 6. During the rebuttal period, Reviewer h1SL explicitly stated in a public comment that the rating was raised from 6 to 8. Although ICLR later globally reverted all scores due to the platform incident, the reviewers’ written comments clearly indicate their intended post-rebuttal evaluations. For Reviewer 1DV6, the three major concerns are on code generation results, additional base RFT methods (such as Dr.GRPO and DAPO) and hyperparameter analysis, respectively. Our additional results addressed the first one, third one, and partially addressed the second one with our results for Dr.GRPO. Since Reviewer 1DV6 is waiting for our results for DAPO, we have reported the first-step results for DAPO in our latest response while more experiments are in progress. **For Reviewer cESh and pHkv, although they have not yet responded in time, we believe our rebuttal and additional experimental results can address all their concerns.** As a result, all reviewers expressed positive positions toward the paper with the post-rebuttal ratings 6, 6, 6, 8 (and could be higher if the discussion stage went normally).

All comments and responses related to the rebuttal, including the subsequent score improvements, were completed BEFORE the OpenReview information-leak incident. Both the reviewer comments and the authors’ responses carry timestamps that explicitly show all discussions, revisions, and score updates occurred prior to the event. No part of the rebuttal or the reviewers’ updated evaluations involved external or unauthorized information.

We greatly appreciate the reviewers’ valuable feedback, which drives us to conduct more experiments that strengthens the completeness of the paper. With all concerns addressed and consistently positive reviewer support, we hope the paper will receive favorable consideration.

---

### Meta-Review · Area_Chair_3fNH · 2026-01-04

**Summary:**

The paper addresses a highly important issue in RFT, rollout inefficiency, and proposes a coherent, modular solution comprising mix-policy reuse, a KL-convex constraint, and a reincarnation handoff. The approach is empirically validated through ablations and efficiency curves. The rebuttal substantially strengthened the submission by broadening the evaluation to code generation, adding the missing 7B ReMix-GRPO results, reporting pass@k (8/16) in addition to pass@1, clarifying theoretical derivations, and providing a sensitivity analysis of the reincarnation trigger. Based on the reviewer questions and the rebuttal, the key concerns raised by all reviewers have been largely addressed.

**Reviewer Concerns:**

1. The evaluation has limited domain/scale/coverage, for example, it is conducted only on math reasoning tasks (cESh, 1DV6), heavily focuses on 7B models (cESh), and lacks ReMix-GRPO results on 7B models (cESh).

2. Some baselines are missing, such as DAPO (1DV6) and off-policy methods (pHkv).

3. Hyperparameter stability, for example, the trigger step T (cESh, 1DV6, h1SL) and the decay schedule (cESh, h1SL).

4. The metrics do not reflect the complete picture, for example, pass@k is missing (h1SL), and on AIME, ReMix-GRPO performs worse than the baseline (h1SL).

**Reviewer Scores:**

Reviewer h1SL: Very likely to raise the score to 7 or 8,
due to the authors adding/clarifying: (i) the missing derivation of Eq. 6, (ii) the distribution formulation, (iii) a sensitivity sweep of the policy reincarnation trigger, (iv) pass@8/16 results, (v) Dr.GRPO experiments and length normalization details, and (vi) a cross-family LLaMA-8B result. In addition, they corrected the Minerva pass@8/16 discrepancy by aligning rollout and evaluation settings.

Reviewer cESh: Very likely to raise the score to 7.
The main concerns were the evaluation scope and missing baselines. During the rebuttal, the authors added code generation results on 1.5B/7B models (LiveCodeBench), added 7B ReMix-GRPO results, and added trigger-step ablation experiments (T = 25/50/100/200).

Reviewer pHkv: Likely to maintain the current score.
The concerns mainly involved clarification of details, including (i) why an adaptive trust region ratio is used in Eq. (3), and (ii) whether early sample reuse / UTD causes instability or bias accumulation. The authors provided explanations for these points.

Reviewer 1DV6: Likely to maintain the score or raise it to 7.
The main concern was the lack of solid benchmarking and baselines. The authors added code generation results on 1.5B/7B models (LiveCodeBench), added a Dr.GRPO baseline and a partial DAPO baseline, and added trigger-step ablation experiments (T = 25/50/100/200).

---

### Decision · Program_Chairs · 2026-01-26

Accept (Poster)